# Adaptive Step–Size for Policy Gradient Methods

**Matteo Pirotta**
Dept. Elect., Inf., and Bio.
Politecnico di Milano, ITALY
matteo.pirotta@polimi.it

**Marcello Restelli**
Dept. Elect., Inf., and Bio.
Politecnico di Milano, ITALY
marcello.restelli@polimi.it

**Luca Bascetta**
Dept. Elect., Inf., and Bio.
Politecnico di Milano, ITALY
luca.bascetta@polimi.it

## Abstract

In the last decade, policy gradient methods have significantly grown in popularity in the reinforcement–learning field. In particular, they have been largely employed in motor control and robotic applications, thanks to their ability to cope with continuous state and action domains and partial observable problems. Policy gradient researches have been mainly focused on the identification of effective gradient directions and the proposal of efficient estimation algorithms. Nonetheless, the performance of policy gradient methods is determined not only by the gradient direction, since convergence properties are strongly influenced by the choice of the step size: small values imply slow convergence rate, while large values may lead to oscillations or even divergence of the policy parameters. Step–size value is usually chosen by hand tuning and still little attention has been paid to its automatic selection. In this paper, we propose to determine the learning rate by maximizing a lower bound to the expected performance gain. Focusing on Gaussian policies, we derive a lower bound that is second–order polynomial of the step size, and we show how a simplified version of such lower bound can be maximized when the gradient is estimated from trajectory samples. The properties of the proposed approach are empirically evaluated in a linear–quadratic regulator problem.

## 1   Introduction

Policy gradient methods have established as the most effective reinforcement–learning techniques in robotic applications. Such methods perform a policy search to maximize the expected return of a policy in a parameterized policy class. The reasons for their success are many. Compared to several traditional reinforcement–learning approaches, policy gradients scale well to high–dimensional continuous state and action problems, and no changes to the algorithms are needed to face uncertainty in the state due to limited and noisy sensors. Furthermore, policy representation can be properly designed for the given task, thus allowing to incorporate domain knowledge into the algorithm useful to speed up the learning process and to prevent the unexpected execution of dangerous policies that may harm the system. Finally, they are guaranteed to converge to locally optimal policies.

Thanks to these advantages, from the 1990s policy gradient methods have been widely used to learn complex control tasks [1]. The research in these years has focused on obtaining good model–free estimators of the policy gradient using data generated during the task execution. The oldest policy gradient approaches are finite–difference methods [2], that estimate gradient direction by resolving a regression problem based on the performance evaluation of policies associated to different small perturbations of the current parameterization. Finite–difference methods have some advantages: they are easy to implement, do not need assumptions on the differentiability of the policy w.r.t. the policy parameters, and are efficient in deterministic settings. On the other hand, when used on real systems, the choice of parameter perturbations may be difficult and critical for system safeness. Furthermore, the presence of uncertainties may significantly slow down the convergence rate. Such drawbacks have been overcome by likelihood ratio methods [3, 4, 5], since they do not need to generate policy parameters variations and quickly converge even in highly stochastic systems. Several

studies have addressed the problem to find minimum variance estimators by the computation of optimal baselines [6]. To further improve the efficiency of policy gradient methods, natural gradient approaches (where the steepest ascent is computed w.r.t. the Fisher information metric) have been considered [7, 8]. Natural gradients still converge to locally optimal policies, are independent from the policy parameterization, need less data to attain good gradient estimate, and are less affected by plateaus.

Once an accurate estimate of the gradient direction is obtained, policy parameters are updated by: $\boldsymbol{\theta}_{t+1} = \boldsymbol{\theta}_t + \alpha_t \nabla_{\boldsymbol{\theta}} J|_{\boldsymbol{\theta}=\boldsymbol{\theta}_t}$, where $\alpha_t \in \mathbb{R}^+$ is the step size in the direction of the gradient. Although, given an unbiased gradient estimate, convergence to a local optimum can be guaranteed under mild conditions over the learning–rate values [9], their choice may significantly affect the convergence speed or the behavior during the transient. Updating the policy with large step sizes may lead to policy oscillations or even divergence [10], while trying to avoid such phenomena by using small learning rates determines a growth in the number of iterations that is unbearable in most real–world applications. In general unconstrained programming, the optimal step size for gradient ascent methods is determined through line–search algorithms [11], that require to try different values for the learning rate and evaluate the function value in the corresponding updated points. Such an approach is unfeasible for policy gradient methods, since it would require to perform a large number of policy evaluations. Despite these difficulties, up to now, little attention has been paid to the study of step–size computation for policy gradient algorithms. Nonetheless, some policy search methods based on expectation–maximization have been recently proposed; such methods have properties similar to the ones of policy gradients, but the policy update does not require to tune the step size [12, 13].

In this paper, we propose a new approach to compute the step size in policy gradient methods that guarantees an improvement at each step, thus avoiding oscillation and divergence issues. Starting from a lower bound to the difference of performance between two policies, in Section 3 we derive a lower bound in the case where the new policy is obtained from the old one by changing its parameters along the gradient direction. Such a new bound is a (polynomial) function of the step size, that, for positive values of the step size, presents a single, positive maximum ( i.e., it guarantees improvement) which can be computed in closed form. In Section 4, we show how the bound simplifies to a quadratic function of the step size when Gaussian policies are considered, and Section 5 studies how the bound needs to be changed in approximated settings (e.g., model–free case) where the policy gradient needs to be estimated directly from experience.

## 2 Preliminaries

A discrete–time continuous Markov decision process (MDP) is defined as a 6-tuple $\langle \mathcal{S}, \mathcal{A}, \mathcal{P}, \mathcal{R}, \gamma, D \rangle$, where $\mathcal{S}$ is the continuous state space, $\mathcal{A}$ is the continuous action space, $\mathcal{P}$ is a Markovian transition model where $\mathcal{P}(s'|s, a)$ defines the transition density between state $s$ and $s'$ under action $a$, $\mathcal{R} : \mathcal{S} \times \mathcal{A} \rightarrow [0, R]$ is the reward function, such that $\mathcal{R}(s, a)$ is the expected immediate reward for the state-action pair $(s, a)$ and $R$ is the maximum reward value, $\gamma \in [0, 1)$ is the discount factor for future rewards, and $D$ is the initial state distribution. The policy of an agent is characterized by a density distribution $\pi(\cdot|s)$ that specifies for each state $s$ the density distribution over the action space $\mathcal{A}$. To measure the distance between two policies we will use this norm:

$$\|\pi' - \pi\|_{\infty} = \sup_{s \in \mathcal{S}} \int_{\mathcal{A}} |\pi'(a|s) - \pi(a|s)| \mathrm{d}a,$$

that is the superior value over the state space of the total variation between the distributions over the action space of policy $\pi'$ and $\pi$.

We consider infinite horizon problems where the future rewards are exponentially discounted with $\gamma$. For each state $s$, we define the utility of following a stationary policy $\pi$ as:

$$V^{\pi}(s) = E_{\substack{a_t \sim \pi \\ s_t \sim \mathcal{P}}} \left[ \sum_{t=0}^{\infty} \gamma^t \mathcal{R}(s_t, a_t) | s_0 = s \right].$$

It is known that $V^{\pi}$ solves the following recursive (Bellman) equation:

$$V^{\pi}(s) = \int_{\mathcal{A}} \pi(a|s) \mathcal{R}(s, a) + \gamma \int_{\mathcal{S}} P(s'|s, a) V^{\pi}(s') \mathrm{d}s' \mathrm{d}a.$$

Policies can be ranked by their expected discounted reward starting from the state distribution $D$:

$$J_D^\pi = \int_\mathcal{S} D(s) V^\pi(s) \mathrm{d}s = \int_\mathcal{S} d_D^\pi(s) \int_\mathcal{A} \pi(a|s) \mathcal{R}(s,a) \mathrm{d}a \mathrm{d}s,$$

where $d_D^\pi(s) = (1-\gamma) \sum_{t=0}^\infty \gamma^t Pr(s_t = s|\pi, D)$ is the $\gamma$–discounted future state distribution for a starting state distribution $D$ [5]. Solving an MDP means to find a policy $\pi^*$ that maximizes the expected long-term reward: $\pi^* \in arg\max_{\pi \in \Pi} J_D^\pi$. For any MDP there exists at least one deterministic optimal policy that simultaneously maximizes $V^\pi(s), \forall s \in \mathcal{S}$. For control purposes, it is better to consider action values $Q^\pi(s,a)$, i.e., the value of taking action $a$ in state $s$ and following a policy $\pi$ thereafter:

$$Q^\pi(s,a) = \mathcal{R}(s,a) + \gamma \int_\mathcal{S} \mathcal{P}(s'|s,a) \int_\mathcal{A} \pi(a'|s') Q^\pi(s',a') \mathrm{d}a' \mathrm{d}s'.$$

Furthermore, we define the advantage function:

$$A^\pi(s,a) = Q^\pi(s,a) - V^\pi(s),$$

that quantifies the advantage (or disadvantage) of taking action $a$ in state $s$ instead of following policy $\pi$. In particular, for each state $s$, we define the advantage of a policy $\pi'$ over policy $\pi$ as $A_\pi^{\pi'}(s) = \int_\mathcal{A} \pi'(a|s) A^\pi(s,a) \mathrm{d}a$ and, following [14], we define its expected value w.r.t. an initial state distribution $\mu$ as $\mathbb{A}_{\pi,\mu}^{\pi'} = \int_\mathcal{S} d_\mu^\pi(s) A_\pi^{\pi'}(s) \mathrm{d}s$.

We consider the problem of finding a policy that maximizes the expected discounted reward over a class of parameterized policies $\Pi_{\boldsymbol{\theta}} = \{\pi_{\boldsymbol{\theta}} : \boldsymbol{\theta} \in \mathbb{R}^m\}$, where $\pi_{\boldsymbol{\theta}}$ is a compact representation of $\pi(a|s, \boldsymbol{\theta})$. The exact gradient of the expected discounted reward w.r.t. the policy parameters [5] is:

$$\nabla_{\boldsymbol{\theta}} J_\mu(\boldsymbol{\theta}) = \frac{1}{1-\gamma} \int_\mathcal{S} d_\mu^{\pi_{\boldsymbol{\theta}}}(s) \int_\mathcal{A} \nabla_{\boldsymbol{\theta}} \pi(a|s, \boldsymbol{\theta}) Q^{\pi_{\boldsymbol{\theta}}}(s,a) \mathrm{d}a \mathrm{d}s.$$

The policy parameters can be updated by following the direction of the gradient of the expected discounted reward: $\boldsymbol{\theta}' = \boldsymbol{\theta} + \alpha \nabla_{\boldsymbol{\theta}} J_\mu(\boldsymbol{\theta})$. In the following, we will denote with $\|\nabla_{\boldsymbol{\theta}} J_\mu(\boldsymbol{\theta})\|_1$ and $\|\nabla_{\boldsymbol{\theta}} J_\mu(\boldsymbol{\theta})\|_2$ the L1– and L2–norm of the policy gradient vector, respectively.

## 3 Policy Gradient Formulation

In this section we provide a lower bound to the improvement obtained by updating the policy parameters along the gradient direction as a function of the step size. The idea is to start from the general lower bound on the performance difference between any pair of policies introduced in [15] and specialize it to the policy gradient framework.

**Lemma 3.1** (Continuous MDP version of Corollary 3.6 in [15]). *For any pair of stationary policies corresponding to parameters $\boldsymbol{\theta}$ and $\boldsymbol{\theta}'$ and for any starting state distribution $\mu$, the difference between the performance of policy $\pi_{\boldsymbol{\theta}'}$ and policy $\pi_{\boldsymbol{\theta}}$ can be bounded as follows*

$$J_\mu(\boldsymbol{\theta}') - J_\mu(\boldsymbol{\theta}) \geq \frac{1}{1-\gamma} \int_\mathcal{S} d_\mu^{\pi_{\boldsymbol{\theta}}}(s) A_{\pi_{\boldsymbol{\theta}}}^{\pi_{\boldsymbol{\theta}'}}(s) \mathrm{d}s - \frac{\gamma}{2(1-\gamma)^2} \|\pi_{\boldsymbol{\theta}'} - \pi_{\boldsymbol{\theta}}\|_\infty^2 \|Q^{\pi_{\boldsymbol{\theta}}}\|_\infty, \quad (1)$$

*where $\|Q^{\pi_{\boldsymbol{\theta}}}\|_\infty$ is the supremum norm of the Q–function: $\|Q^{\pi_{\boldsymbol{\theta}}}\|_\infty = \sup\limits_{s \in \mathcal{S}, a \in \mathcal{A}} Q^{\pi_{\boldsymbol{\theta}}}(s,a)$*

As we can notice from the above bound, to maximize the performance improvement, we need to find a new policy $\pi_{\boldsymbol{\theta}'}$ that is associated to large average advantage $\mathbb{A}_{\pi_{\boldsymbol{\theta}},\mu}^{\pi_{\boldsymbol{\theta}'}}$, but, at the same time, is not too different from the current policy $\pi_{\boldsymbol{\theta}}$. Policy gradient approaches provide search directions characterized by increasing advantage values and, through the step size value, allow to control the difference between the new policy and the target one. Exploiting a lower bound to the first order Taylor's expansion, we can bound the difference between the current policy and the new policy, whose parameters are adjusted along the gradient direction, as a function of the step size $\alpha$.

**Lemma 3.2.** *Let the update of the policy parameters be $\boldsymbol{\theta}' = \boldsymbol{\theta} + \alpha \nabla_{\boldsymbol{\theta}} J_\mu(\boldsymbol{\theta})$. Then*

$$\pi(a|s, \boldsymbol{\theta}') - \pi(a|s, \boldsymbol{\theta}) \geq \alpha \nabla_{\boldsymbol{\theta}} \pi(a|s, \boldsymbol{\theta})^\top \nabla_{\boldsymbol{\theta}} J_\mu(\boldsymbol{\theta}) + \alpha^2 \inf_{c \in (0,1)} \left( \sum_{i,j=1}^m \left. \frac{\partial^2 \pi(a|s, \boldsymbol{\theta})}{\partial \theta_i \partial \theta_j} \right|_{\boldsymbol{\theta}+c\Delta\boldsymbol{\theta}} \frac{\Delta\theta_i \Delta\theta_j}{1 + I(i=j)} \right),$$

*where $\Delta\boldsymbol{\theta} = \alpha \nabla_{\boldsymbol{\theta}} J_\mu(\boldsymbol{\theta})$.*

By combining the two previous lemmas, it is possible to derive the policy performance improvement obtained following the gradient direction.

**Theorem 3.3.** *Let the update of the parameters be $\boldsymbol{\theta}' = \boldsymbol{\theta} + \alpha\nabla_{\boldsymbol{\theta}}J_\mu(\boldsymbol{\theta})$. Then for any stationary policy $\pi(a|s, \boldsymbol{\theta})$ and any starting state distribution $\mu$, the difference in performance between $\pi_{\boldsymbol{\theta}}$ and $\pi_{\boldsymbol{\theta}'}$ is lower bounded by:*

$$
J_\mu(\boldsymbol{\theta}') - J_\mu(\boldsymbol{\theta}) \geq \alpha\left\|\nabla_{\boldsymbol{\theta}}J_\mu(\boldsymbol{\theta})\right\|_2^2
$$
$$
+ \frac{\alpha^2}{1-\gamma}\int_{\mathcal{S}}d_\mu^{\pi_{\boldsymbol{\theta}}}(s)\int_{\mathcal{A}}\inf_{c\in(0,1)}\left(\sum_{i,j=1}^{m}\left.\frac{\partial^2\pi(a|s,\boldsymbol{\theta})}{\partial\theta_i\partial\theta_j}\right|_{\boldsymbol{\theta}+c\Delta\boldsymbol{\theta}}\frac{\Delta\theta_i\,\Delta\theta_j}{1+I(i=j)}\right)Q^{\pi_{\boldsymbol{\theta}}}(s,a)\mathrm{d}a\mathrm{d}s
$$
$$
- \frac{\gamma\left\|Q^{\pi_{\boldsymbol{\theta}}}\right\|_\infty}{2(1-\gamma)^2}\left(\alpha\sup_{s\in\mathcal{S}}\int_{\mathcal{A}}\left|\nabla_{\boldsymbol{\theta}}\pi(a|s,\boldsymbol{\theta})^T\nabla_{\boldsymbol{\theta}}J_\mu(\boldsymbol{\theta})\right|\mathrm{d}a\right.
$$
$$
\left.+\alpha^2\sup_{s\in\mathcal{S}}\int_{\mathcal{A}}\left|\sup_{c\in(0,1)}\left(\sum_{i,j=1}^{m}\left.\frac{\partial^2\pi(a|s,\boldsymbol{\theta})}{\partial\theta_i\partial\theta_j}\right|_{\boldsymbol{\theta}+c\Delta\boldsymbol{\theta}}\frac{\Delta\theta_i\,\Delta\theta_j}{1+I(i=j)}\right)\right|\mathrm{d}a\right)^2.
$$

The above bound is a forth–order polynomial of the step size, whose stationary points, being the roots of a third–order polynomial $ax^3 + bx^2 + cx + d$, can be expressed in closed form. It is worth to notice that, for positive values of $\alpha$, the bound presents a single stationary point that corresponds to a local maximum. In fact, since $a, b \leq 0$ and $d \geq 0$, the Descartes' rule of signs gives the existence and uniqueness of the real positive root.

In the following section, we will show, in the case of Gaussian policies, how the bound in Theorem 3.3 can be reduced to a second–order polynomial in $\alpha$, thus obtaining a simpler closed-form solution for optimal (w.r.t. the bound) step size.

## 4   The Gaussian Policy Model

In this section we consider the Gaussian policy model with fixed standard deviation $\sigma$ and the mean is a linear combination of the state feature vector $\phi(\cdot)$ using a parameter vector $\boldsymbol{\theta}$ of size $m$:

$$
\pi(a|s,\boldsymbol{\theta}) = \frac{1}{\sqrt{2\pi\sigma^2}}\exp\left(-\frac{1}{2}\left(\frac{a-\boldsymbol{\theta}^{\mathrm{T}}\phi(s)}{\sigma}\right)^2\right).
$$

In the case of Gaussian policies, each second–order derivative of policy $\pi_{\boldsymbol{\theta}}$ can be easily bounded.

**Lemma 4.1.** *For any Gaussian policy $\pi(a|s,\boldsymbol{\theta}) \sim \mathcal{N}(\boldsymbol{\theta}^T\phi(s), \sigma^2)$, the second order derivative of the policy can be bounded as follows:*

$$
\left|\frac{\partial^2\pi(a|s,\boldsymbol{\theta})}{\partial\theta_i\partial\theta_j}\right| \leq \frac{|\phi_i(s)\phi_j(s)|}{\sqrt{2\pi}\sigma^3}, \quad \forall\boldsymbol{\theta}\in\mathbb{R}^m, \forall a\in\mathcal{A}.
$$

*This result allows to restate Lemma 3.2 in the case of Gaussian policies:*

$$
\pi(a|s,\boldsymbol{\theta}') - \pi(a|s,\boldsymbol{\theta}) \geq \alpha\nabla_{\boldsymbol{\theta}}\pi(a|s,\boldsymbol{\theta})^T\nabla_{\boldsymbol{\theta}}J_\mu(\boldsymbol{\theta}) - \frac{\alpha^2}{\sqrt{2\pi}\sigma^3}\left(|\nabla_{\boldsymbol{\theta}}J_\mu(\boldsymbol{\theta})|^T|\phi(s)|\right)^2.
$$

In the following we will assume that features $\phi$ are uniformly bounded:

**Assumption 4.1.** All the basis functions are uniformly bounded by $M_\phi$: $|\phi_i(s)| < M_\phi$, $\forall s \in \mathcal{S}, \forall i = 1,\ldots,m$.

Exploiting Pinsker's inequality [16] (which upper bounds the total variation between two distributions with their Kullback–Liebler divergence), it is possible to provide the following upper bound to the supremum norm between two Gaussian policies.

**Lemma 4.2.** *For any pair of stationary policies $\pi_{\boldsymbol{\theta}}$ and $\pi_{\boldsymbol{\theta}'}$, so that $\boldsymbol{\theta}' = \boldsymbol{\theta} + \alpha\nabla_{\boldsymbol{\theta}}J_\mu(\boldsymbol{\theta})$, supremum norm of their difference can be upper bounded as follows:*

$$
\left\|\pi_{\boldsymbol{\theta}'} - \pi_{\boldsymbol{\theta}}\right\|_\infty \leq \frac{\alpha M_\phi}{\sigma}\left\|\nabla_{\boldsymbol{\theta}}J_\mu(\boldsymbol{\theta})\right\|_1.
$$

By plugging the results of Lemmas 4.1 and 4.2 into Equation (1) we can obtain a lower bound to the performance difference between a Gaussian policy $\pi_{\boldsymbol{\theta}}$ and another policy along the gradient direction that is quadratic in the step size $\alpha$.

**Theorem 4.3.** *For any starting state distribution $\mu$, and any pair of stationary Gaussian policies $\pi_{\boldsymbol{\theta}} \sim \mathcal{N}(\boldsymbol{\theta}^T \boldsymbol{\phi}(s), \sigma^2)$ and $\pi_{\boldsymbol{\theta}'} \sim \mathcal{N}(\boldsymbol{\theta'}^T \boldsymbol{\phi}(s), \sigma^2)$, so that $\boldsymbol{\theta}' = \boldsymbol{\theta} + \alpha \nabla_{\boldsymbol{\theta}} J_\mu(\boldsymbol{\theta})$ and under Assumption 4.1, the difference between the performance of $\pi_{\boldsymbol{\theta}'}$ and the one of $\pi_{\boldsymbol{\theta}}$ can be lower bounded as follows:*

$$
\begin{aligned}
J_\mu(\boldsymbol{\theta}') - J_\mu(\boldsymbol{\theta}) \geq{}& \alpha \left\| \nabla_{\boldsymbol{\theta}} J_\mu(\boldsymbol{\theta}) \right\|_2^2 \\
& - \alpha^2 \left( \frac{1}{(1-\gamma)\sqrt{2\pi}\sigma^3} \int_{\mathcal{S}} d_\mu^{\pi_{\boldsymbol{\theta}}}(s) \left( |\nabla_{\boldsymbol{\theta}} J_\mu(\boldsymbol{\theta})|^T |\boldsymbol{\phi}(s)| \right)^2 \int_{\mathcal{A}} Q^{\pi_{\boldsymbol{\theta}}}(s,a) \mathrm{d}a \mathrm{d}s \right. \\
& \left. + \frac{\gamma M_\phi^2}{2(1-\gamma)^2 \sigma^2} \left\| \nabla_{\boldsymbol{\theta}} J_\mu(\boldsymbol{\theta}) \right\|_1^2 \left\| Q^{\pi_{\boldsymbol{\theta}}} \right\|_\infty \right).
\end{aligned}
$$

Since the linear coefficient is positive and the quadratic one is negative, the bound in Theorem 4.3 has a single maximum attained for some positive value of $\alpha$.

**Corollary 4.4.** *The performance lower bound provided in Theorem 4.3 is maximized by choosing the following step size:*

$$
\alpha^* = \frac{(1-\gamma)^2 \sqrt{2\pi}\sigma^3 \left\| \nabla_{\boldsymbol{\theta}} J_\mu(\boldsymbol{\theta}) \right\|_2^2}{\gamma\sqrt{2\pi}\sigma M_\phi^2 \left\| \nabla_{\boldsymbol{\theta}} J_\mu(\boldsymbol{\theta}) \right\|_1^2 \left\| Q^{\pi_{\boldsymbol{\theta}}} \right\|_\infty + 2(1-\gamma) \int_{\mathcal{S}} d_\mu^{\pi_{\boldsymbol{\theta}}}(s) \left( |\nabla_{\boldsymbol{\theta}} J_\mu(\boldsymbol{\theta})|^T |\boldsymbol{\phi}(s)| \right)^2 \int_{\mathcal{A}} Q^{\pi_{\boldsymbol{\theta}}}(s,a) \mathrm{d}a \mathrm{d}s},
$$

*that guarantees the following policy performance improvement*

$$
J_\mu(\boldsymbol{\theta}') - J_\mu(\boldsymbol{\theta}) \geq \frac{1}{2}\alpha^* \left\| \nabla_{\boldsymbol{\theta}} J_\mu(\boldsymbol{\theta}) \right\|_2^2.
$$

# 5   Approximate Framework

The solution for the tuning of the step size presented in the previous section depends on some constants (e.g., discount factor and the variance of the Gaussian policy) and requires to be able to compute some quantities (e.g., the policy gradient and the supremum value of the $Q$–function). In many real–world applications such quantities cannot be computed (e.g., when the state–transition model is unknown or too large for exact methods) and need to be estimated from experience samples. In this section, we study how the step size can be chosen when the gradient is estimated through sample trajectories to guarantee a performance improvement in high probability.

For sake of easiness, we consider a simplified version of the bound in Theorem 4.3, in order to obtain a bound where the only element that needs to be estimated is the policy gradient $\nabla_{\boldsymbol{\theta}} J_\mu(\boldsymbol{\theta})$.

**Corollary 5.1.** *For any starting state distribution $\mu$, and any pair of stationary Gaussian policies $\pi_{\boldsymbol{\theta}} \sim \mathcal{N}(\boldsymbol{\theta}^T \boldsymbol{\phi}(s), \sigma^2)$ and $\pi_{\boldsymbol{\theta}'} \sim \mathcal{N}(\boldsymbol{\theta'}^T \boldsymbol{\phi}(s), \sigma^2)$, so that $\boldsymbol{\theta}' = \boldsymbol{\theta} + \alpha \nabla_{\boldsymbol{\theta}} J_\mu(\boldsymbol{\theta})$ and under Assumption 4.1, the difference between the performance of $\pi_{\boldsymbol{\theta}'}$ and $\pi_{\boldsymbol{\theta}}$ is lower bounded by:*

$$
J_\mu(\boldsymbol{\theta}') - J_\mu(\boldsymbol{\theta}) \geq \alpha \left\| \nabla_{\boldsymbol{\theta}} J_\mu(\boldsymbol{\theta}) \right\|_2^2 - \alpha^2 \frac{R M_\phi^2 \left\| \nabla_{\boldsymbol{\theta}} J_\mu(\boldsymbol{\theta}) \right\|_1^2}{(1-\gamma)^2 \sigma^2} \left( \frac{|\mathcal{A}|}{\sqrt{2\pi}\sigma} + \frac{\gamma}{2(1-\gamma)} \right),
$$

*that is maximized by the following step size value:*

$$
\tilde{\alpha}^* = \frac{(1-\gamma)^3 \sqrt{2\pi}\sigma^3 \left\| \nabla_{\boldsymbol{\theta}} J_\mu(\boldsymbol{\theta}) \right\|_2^2}{\left( \gamma\sqrt{2\pi}\sigma + 2(1-\gamma)|\mathcal{A}| \right) R M_\phi^2 \left\| \nabla_{\boldsymbol{\theta}} J_\mu(\boldsymbol{\theta}) \right\|_1^2}.
$$

Since we are assuming that the policy gradient $\nabla_{\boldsymbol{\theta}} J_\mu(\boldsymbol{\theta})$ is estimated through trajectory samples, the lower bound in Corollary 5.1 must take into consideration the associated approximation error. Given a set of trajectories obtained following policy $\pi_{\boldsymbol{\theta}}$, we can produce an estimate $\widehat{\nabla}_{\boldsymbol{\theta}} J_\mu(\boldsymbol{\theta})$ of the policy gradient and we assume to be able to produce a vector $\boldsymbol{\epsilon} = [\epsilon_1, \ldots, \epsilon_m]^T$, so that the $i$–th component of the approximation error is bounded at least with probability $1 - \delta$:

$$
\mathbb{P}\left( \left| \nabla_{\theta_i} J_\mu(\boldsymbol{\theta}) - \widehat{\nabla}_{\theta_i} J_\mu(\boldsymbol{\theta}) \right| \geq \epsilon_i \right) \leq \delta.
$$

Given the approximation error vector $\epsilon$, we can adjust the bound in Corollary 5.1 to produce a new bound that holds at least with probability $(1 - \delta)^m$. In particular, to preserve the inequality sign, the estimated approximation error must be used to decrease the L2–norm of the policy gradient in the first term (the one that provides the positive contribution to the performance improvement) and to increase the L1–norm in the penalization term. To lower bound the L2–norm, we introduce the vector $\underline{\widehat{\nabla}_{\boldsymbol{\theta}} J_\mu}(\boldsymbol{\theta})$ whose components are a lower bound to the absolute value of the policy gradient built on the basis of the approximation error $\epsilon$:

$$\underline{\widehat{\nabla}_{\boldsymbol{\theta}} J_\mu}(\boldsymbol{\theta}) = \max(|\widehat{\nabla}_{\boldsymbol{\theta}} J_\mu(\boldsymbol{\theta})| - \boldsymbol{\epsilon}, \mathbf{0}),$$

where $\mathbf{0}$ denotes the $m$–size vector with all zeros, and $\max$ denotes the component–wise maximum. Similarly, to upper bound the L1–norm of the policy gradient, we introduce the vector $\overline{\widehat{\nabla}_{\boldsymbol{\theta}} J_\mu}(\boldsymbol{\theta})$:

$$\overline{\widehat{\nabla}_{\boldsymbol{\theta}} J_\mu}(\boldsymbol{\theta}) = |\widehat{\nabla}_{\boldsymbol{\theta}} J_\mu(\boldsymbol{\theta})| + \boldsymbol{\epsilon}.$$

**Theorem 5.2.** *Under the same assumptions of Corollary 5.1, and provided that it is available a policy gradient estimate $\widehat{\nabla}_{\boldsymbol{\theta}} J_\mu(\boldsymbol{\theta})$, so that $\mathbb{P}\left( \left| \nabla_{\theta_i} J_\mu(\boldsymbol{\theta}) - \widehat{\nabla}_{\theta_i} J_\mu(\boldsymbol{\theta}) \right| \geq \epsilon_i \right) \leq \delta$, the difference between the performance of $\pi_{\boldsymbol{\theta}'}$ and $\pi_{\boldsymbol{\theta}}$ can be lower bounded at least with probability $(1 - \delta)^m$:*

$$J_\mu(\boldsymbol{\theta}') - J_\mu(\boldsymbol{\theta}) \geq \alpha \left\| \underline{\widehat{\nabla}_{\boldsymbol{\theta}} J_\mu}(\boldsymbol{\theta}) \right\|_2^2 - \alpha^2 \frac{R M_\phi^2 \left\| \overline{\widehat{\nabla}_{\boldsymbol{\theta}} J_\mu}(\boldsymbol{\theta}) \right\|_1^2}{(1 - \gamma)^2 \sigma^2} \left( \frac{|\mathcal{A}|}{\sqrt{2\pi}\sigma} + \frac{\gamma}{2(1 - \gamma)} \right),$$

*that is maximized by the following step size value:*

$$\widehat{\alpha}^* = \frac{(1 - \gamma)^3 \sqrt{2\pi}\sigma^3 \left\| \underline{\widehat{\nabla}_{\boldsymbol{\theta}} J_\mu}(\boldsymbol{\theta}) \right\|_2^2}{\left( \gamma\sqrt{2\pi}\sigma + 2(1 - \gamma)|\mathcal{A}| \right) R M_\phi^2 \left\| \overline{\widehat{\nabla}_{\boldsymbol{\theta}} J_\mu}(\boldsymbol{\theta}) \right\|_1^2}.$$

In the following, we will discuss how the approximation error of the policy gradient can be bounded. Among the several methods that have been proposed over the years, we focus on two well–understood policy–gradient estimation approaches: REINFORCE [3] and G(PO)MDP [4]/policy gradient theorem (PGT) [5].

## 5.1 Approximation with REINFORCE gradient estimator

The REINFORCE approach [3] is the main exponent of the likelihood–ratio family. The episodic REINFORCE gradient estimator is given by:

$$\widehat{\nabla}_{\boldsymbol{\theta}} J_\mu^{RF}(\boldsymbol{\theta}) = \frac{1}{N} \sum_{n=1}^{N} \left( \sum_{k=1}^{H} \nabla_{\boldsymbol{\theta}} \log \pi \left( a_k^n; s_k^n, \boldsymbol{\theta} \right) \left( \sum_{l=1}^{H} \gamma^{l-1} r_l^n - b \right) \right),$$

where $N$ is the number of $H$–step trajectories generated from a system by roll–outs and $b \in \mathbb{R}$ is a baseline that can be chosen arbitrary, but usually with the goal of minimizing the variance of the gradient estimator. The main drawback of REINFORCE is its variance, that is strongly affected by the length of the trajectory horizon $H$.

The goal is to determine the number of trajectories $N$ in order to obtain the desired accuracy of the gradient estimate. To achieve this, we exploit the upper bound to the variance of the episodic REINFORCE gradient estimator introduced in [17] for Gaussian policies.

**Lemma 5.3** (Adapted from Theorem 2 in [17]). *Given a Gaussian policy $\pi(a|s, \boldsymbol{\theta}) \sim \mathcal{N}\left( \boldsymbol{\theta}^\top \phi(s), \sigma^2 \right)$, under the assumption of uniformly bounded rewards and basis functions (Assumption 4.1), we have the following upper bound to the variance of the $i$–th component of the episodic REINFORCE gradient estimate $\widehat{\nabla}_{\theta_i} J_\mu^{RF}(\boldsymbol{\theta})$:*

$$Var\left( \widehat{\nabla}_{\theta_i} J_\mu^{RF}(\boldsymbol{\theta}) \right) \leq \frac{R^2 M_\phi^2 H \left( 1 - \gamma^H \right)^2}{N \sigma^2 (1 - \gamma)^2}.$$

The result in the previous Lemma combined with the Chebyshev's inequality allows to provide a high–probability upper bound to the gradient approximation error using the episodic REINFORCE gradient estimator.

**Theorem 5.4.** *Given a Gaussian policy $\pi(a|s, \boldsymbol{\theta}) \sim \mathcal{N}\left(\boldsymbol{\theta}^T \phi(s), \sigma^2\right)$, under the assumption of uniformly bounded rewards and basis functions (Assumption 4.1), using the following number of $H$–step trajectories:*

$$N = \frac{R^2 M_\phi^2 H \left(1 - \gamma^H\right)^2}{\delta \epsilon_i^2 \sigma^2 \left(1 - \gamma\right)^2},$$

*the gradient estimate $\widehat{\nabla}_{\theta_i} J_\mu^{RF}(\boldsymbol{\theta})$ generated by REINFORCE is such that with probability $1 - \delta$:*

$$\left| \widehat{\nabla}_{\theta_i} J_\mu^{RF}(\boldsymbol{\theta}) - \nabla_{\theta_i} J_\mu(\boldsymbol{\theta}) \right| \leq \epsilon_i.$$

## 5.2 Approximation with G(PO)MDP/PGT gradient estimator

Although the REINFORCE method is guaranteed to converge at the true gradient at the fastest possible pace, its large variance can be problematic in practice. Advances in the likelihood ratio gradient estimators have produced new approaches that significantly reduce the variance of the estimate. Focusing on the class of "vanilla" gradient estimator, two main approaches have been proposed: policy gradient theorem (PGT) [5] and G(PO)MDP [4]. In [6], the authors show that, while the algorithms look different, their gradient estimate are equal, i.e., $\widehat{\nabla}_{\boldsymbol{\theta}} J_\mu^{PGT}(\boldsymbol{\theta}) = \widehat{\nabla}_{\boldsymbol{\theta}} J_\mu^{G(PO)MDP}(\boldsymbol{\theta})$. For this reason, we can limit our attention to the PGT formulation:

$$\widehat{\nabla}_{\boldsymbol{\theta}} J_\mu^{PGT}(\boldsymbol{\theta}) = \frac{1}{N} \sum_{n=1}^{H} \left( \sum_{k=1}^{H} \nabla_{\boldsymbol{\theta}} \log \pi \left(a_k^n; s_k^n, \boldsymbol{\theta}\right) \left( \sum_{l=k}^{H} \gamma^{l-1} r_l^n - b_l^n \right) \right),$$

where $b_l^n \in \mathbb{R}$ have the objective to reduce the variance of the gradient estimate. Following the procedure used to bound the approximation error of REINFORCE, we need an upper bound to the variance of the gradient estimate of PGT that is provided by the following lemma (whose proof is similar to the one used in [17] for the REINFORCE case).

**Lemma 5.5.** *Given a Gaussian policy $\pi(a|s, \boldsymbol{\theta}) \sim \mathcal{N}\left(\boldsymbol{\theta}^T \phi(s), \sigma^2\right)$, under the assumption of uniformly bounded rewards and basis functions (Assumption 4.1), we have the following upper bound to the variance of the $i$–th component of the PGT gradient estimate $\widehat{\nabla}_{\theta_i} J_\mu^{PGT}(\boldsymbol{\theta})$:*

$$Var\left( \widehat{\nabla}_{\theta_i} J_\mu^{PGT}(\boldsymbol{\theta}) \right) \leq \frac{R^2 M_\phi^2}{N \left(1 - \gamma\right)^2 \sigma^2} \left[ \frac{1 - \gamma^{2H}}{1 - \gamma^2} + H\gamma^{2H} - 2\gamma^H \frac{1 - \gamma^H}{1 - \gamma} \right].$$

As expected, since the variance of the gradient estimate obtained with PGT is smaller than the one with REINFORCE, also the upper bound of the PGT variance is smaller than REINFORCE one. In particular, while the variance with REINFORCE grows linearly with the time horizon, using PGT the dependence on the time horizon is significantly smaller. Finally, we can derive the upper bound for the approximation error of the gradient estimated of PGT.

**Theorem 5.6.** *Given a Gaussian policy $\pi(a|s, \boldsymbol{\theta}) \sim \mathcal{N}\left(\boldsymbol{\theta}^T \phi(s), \sigma^2\right)$, under the assumption of uniformly bounded rewards and basis functions (Assumption 4.1), using the following number of $H$–step trajectories:*

$$N = \frac{R^2 M_\phi^2}{\delta \epsilon_i^2 \sigma^2 \left(1 - \gamma\right)^2} \left[ \frac{1 - \gamma^{2H}}{1 - \gamma^2} + H\gamma^{2H} - 2\gamma^H \frac{1 - \gamma^H}{1 - \gamma} \right]$$

*the gradient estimate $\widehat{\nabla}_{\theta_i} J_\mu^{PGT}(\boldsymbol{\theta})$ generated by PGT is such that with probability $1 - \delta$:*

$$\left| \widehat{\nabla}_{\theta_i} J_\mu^{PGT}(\boldsymbol{\theta}) - \nabla_{\theta_i} J_\mu(\boldsymbol{\theta}) \right| \leq \epsilon_i.$$

|  |  | σ | | | | | | | | |
|---|---|---|---|---|---|---|---|---|---|---|
|  |  | 0.50 | 0.75 | 1.00 | 1.25 | 1.50 | 1.75 | 2.00 | 5.00 | 7.50 |
| $\alpha_{const}$ | $1e-07$ | it$_{max}$ | it$_{max}$ | it$_{max}$ | it$_{max}$ | it$_{max}$ | it$_{max}$ | it$_{max}$ | 21888 | 9740 |
|  | $1e-06$ | it$_{max}$ | it$_{max}$ | it$_{max}$ | it$_{max}$ | 23651 | 17516 | 13480 | 2163 | 849 |
|  | $1e-05$ | 17138 | 8669 | 5120 | 3348 | 2342 | 1714 | 1287 | ⊥ | ⊥ |
|  | $1e-04$ | **1675** | **697** | **499** | ⊥ | ⊥ | ⊥ | ⊥ | ⊥ | ⊥ |
|  | $1e-03$ | ⊥ | ⊥ | ⊥ | ⊥ | ⊥ | ⊥ | ⊥ | ⊥ | ⊥ |
| $\alpha_t = \frac{\alpha_0}{t}$ | $1e-05$ | it$_{max}$ | it$_{max}$ | it$_{max}$ | it$_{max}$ | it$_{max}$ | it$_{max}$ | it$_{max}$ | ⊥ | ⊥ |
|  | $1e-04$ | it$_{max}$ | it$_{max}$ | it$_{max}$ | ⊥ | ⊥ | ⊥ | ⊥ | ⊥ | ⊥ |
| $\alpha^*$ |  | 24106 | 7271 | 3279 | **1838** | **1172** | **813** | **598** | **1** | **58** |

Table 1: Convergence speed in exact LQG scenario with $\gamma = 0.95$. The table reports the number of iterations required by the exact gradient approach, starting from $\theta = 0$, to learn the optimal policy parameter $\theta^* = -0.6037$ with an accuracy of $0.01$, for different step–size values. Three different set of experiments are shown: constant step size, decreasing step size, and the step size proposed in Corollary 4.4. The table contains it$_{max}$ when no convergence happens in $30,000$ iterations, and $\perp$ when the algorithm diverges ($\theta < -1$ or $\theta > 0$). Best performances are reported in boldface.

|  | Number of trajectories | | | | | |
|---|---|---|---|---|---|---|
|  | 10,000 | | 100,000 | | 500,000 | |
|  | it | $\theta$ | it | $\theta$ | it | $\theta$ |
| RF | 822 | $-0.0030$ | 51,731 | $-0.3068$ | 75,345 | $-0.4088$ |
| PGT | 29,761 | $-0.2176$ | 63,985 | $-0.4013$ | 83,983 | $-0.4558$ |

Table 2: Convergence speed in approximate LQG scenario with $\gamma = 0.9$. The table reports, starting from $\theta = 0$ and fixed $\sigma = 1$, the number of iterations performed before the proposed step size $\widehat{\alpha}$ becomes $0$ and the last value of the policy parameter. Results are shown for different number of trajectories (of 20 steps each) used in the gradient estimation by REINFORCE and PGT.

## 6 Numerical Simulations and Discussion

In this section we show results related to some numerical simulations of policy gradient in the linear–quadratic Gaussian regulation (LQG) problem as formulated in [6]. The LQG problem is characterized by a transition model $s_{t+1} \sim \mathcal{N}\left(s_t + a_t, \sigma^2\right)$, Gaussian policy $a_t \sim \mathcal{N}\left(\theta \cdot s, \sigma^2\right)$ and quadratic reward $r_t = -0.5(s_t^2 + a_t^2)$. The range of state and action spaces is bounded to the interval $[-2, 2]$ and the initial state is drawn uniformly at random. This scenario is particularly instructive since it allows to exactly compute all terms involved in the bounds. We first present results in the exact scenario and then we move toward the approximated one.

Table 1 shows how the number of iterations required to learn a near–optimal value of the policy parameter changes according to the standard deviation of the Gaussian policy and the step–size value. As expected, very small values of the step size allow to avoid divergence, but the learning process needs many iterations to reach a good performance (this can be observed both when the step size is kept constant and when it decreases). On the other hand, larger step–size values may lead to divergence. In this example, the higher the policy variance, the lower is the step size value that allows to avoid divergence, since, in LQG, higher policy variance implies larger policy gradient values. Using the step size $\alpha^*$ from Corollary 4.4 the policy gradient algorithm avoids divergence (since it guarantees an improvement at each iteration), and the speed of convergence is strongly affected by the variance of the Gaussian policy. In general, when the policy are nearly deterministic (small variance in the Gaussian case), small changes in the parameters lead to large distances between the policies, thus negatively affecting the lower bound in Equation 1. As we can notice from the expression of $\alpha^*$ in Corollary 4.4, considering policies with high variance (that might be a problem in real–world applications) allows to safely take larger step size, thus speeding up the learning process. Nonetheless, increasing the variance over some threshold (making policies nearly random) produces very bad policies, so that changing the policy parameter has a small impact on the performance, and as a result slows down the learning process. How to identify an optimal variance value is an interesting future research direction. Table 2 provides numerical results in the approximated settings, showing the effect of varying the number of trajectories used to estimate the gradient by REINFORCE and PGT. Increasing the number of trajectories reduces the uncertainty on the gradient estimates, thus allowing to use larger step sizes and reaching better performances. Furthermore, the smaller variance of PGT w.r.t. REINFORCE allows the former to achieve better performances. However, even with a large number of trajectories, the approximated errors are still quite large preventing to reach very high performance. For this reason, future studies will try to derive tighter bounds. Further developments include extending these results to other policy models (e.g., Gibbs policies) and to other policy gradient approaches (e.g., natural gradient).

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
