[Supplementary Material]

# Adaptive Step–Size for Policy Gradient Methods – Supplementary Material

**Matteo Pirotta**
Dept. Elect., Inf., and Bioeng.
Politecnico di Milano, Milan, ITALY
`matteo.pirotta@polimi.it`

**Marcello Restelli**
Dept. Elect., Inf., and Bioeng.
Politecnico di Milano, Milan, ITALY
`marcello.restelli@polimi.it`

**Luca Bascetta**
Dept. Elect., Inf., and Bioeng.
Politecnico di Milano, Milan, ITALY
`luca.bascetta@polimi.it`

This document contains supplementary material for the paper "Adaptive Step–Size for Policy Gradient Methods", submitted to the *Neural Information Processing Systems (NIPS) 2013*. It follows the same structure of the main article. For each section we report the complete set of proofs and some additional details. Concerning the numerical simulation, we add some figures that help in understanding the behavior of the proposed approach.

## 1   Policy Gradient Formulation

**Lemma 3.2.** *Let the update of the policy parameters be* $\boldsymbol{\theta}' = \boldsymbol{\theta} + \alpha \nabla_{\boldsymbol{\theta}} J_\mu(\boldsymbol{\theta})$. *Then*

$$
\pi(a|s, \boldsymbol{\theta}') - \pi(a|s, \boldsymbol{\theta}) \geq \alpha \nabla_{\boldsymbol{\theta}} \pi(a|s, \boldsymbol{\theta})^{\mathsf{T}} \nabla_{\boldsymbol{\theta}} J_\mu(\boldsymbol{\theta}) + \alpha^2 \inf_{c \in (0,1)} \left( \sum_{i,j=1}^{m} \left. \frac{\partial^2 \pi(a|s, \boldsymbol{\theta})}{\partial \theta_i \partial \theta_j} \right|_{\boldsymbol{\theta} + c\Delta\boldsymbol{\theta}} \frac{\Delta\theta_i \, \Delta\theta_j}{1 + I(i=j)} \right),
$$

*where* $\Delta\boldsymbol{\theta} = \alpha \nabla_{\boldsymbol{\theta}} J_\mu(\boldsymbol{\theta})$.

*Proof.* The adjustment in the parameter vector $\boldsymbol{\theta}$ is $\Delta\boldsymbol{\theta} = \alpha \nabla_{\boldsymbol{\theta}} J_\mu(\boldsymbol{\theta})$. As a consequence, the first order approximation of the improved policy $\pi(a|s, \boldsymbol{\theta}')$ is:

$$
\begin{aligned}
\pi(a|s, \boldsymbol{\theta}') &= \pi(a|s, \boldsymbol{\theta}) + \nabla_{\boldsymbol{\theta}'} \pi(a|s, \boldsymbol{\theta}')^{\mathsf{T}} \Big|_{\boldsymbol{\theta}} \Delta\boldsymbol{\theta} + R_1(\Delta\boldsymbol{\theta}) \\
&= \pi(a|s, \boldsymbol{\theta}) + \alpha \nabla_{\boldsymbol{\theta}} \pi(a|s, \boldsymbol{\theta})^{\mathsf{T}} \nabla_{\boldsymbol{\theta}} J_\mu(\boldsymbol{\theta}) + R_1(\Delta\boldsymbol{\theta}) \quad \forall s \in \mathcal{S}, \; \forall a \in \mathcal{A}
\end{aligned}
$$

The remainder is given in multi–index Lagrange form by:

$$
R_1(\Delta\boldsymbol{\theta}) = \sum_{|\beta|=2} D^\beta \pi(a|s, \boldsymbol{\theta} + c\Delta\boldsymbol{\theta}) \frac{\Delta\boldsymbol{\theta}^\beta}{\beta!} \qquad \text{for some } c \in (0,1)
$$

where $\beta$ is a multi–index, $|\beta| = \beta_1 + \beta_2 + \cdots + \beta_m$ and $\beta! = \beta_1! \beta_2! \dots \beta_m!$. A lower bound is easily derived by minimizing the remainder along the line connecting the current parameterization $\boldsymbol{\theta}$ and the value $\boldsymbol{\theta} + \Delta\boldsymbol{\theta}$:

$$
\begin{aligned}
R_1(\Delta\boldsymbol{\theta}) &= \sum_{i=1}^{m} \sum_{j=1}^{m} \left. \frac{\partial^2 \pi(a|s, \boldsymbol{\theta})}{\partial \theta_i \partial \theta_j} \right|_{\boldsymbol{\theta} + c\Delta\boldsymbol{\theta}} \frac{\Delta\theta_i \, \Delta\theta_j}{1 + \mathrm{I}(i=j)} \qquad \text{for some } c \in (0,1) \\
&\geq \inf_{c \in (0,1)} \left( \sum_{i=1}^{m} \sum_{j=1}^{m} \left. \frac{\partial^2 \pi(a|s, \boldsymbol{\theta})}{\partial \theta_i \partial \theta_j} \right|_{\boldsymbol{\theta} + c\Delta\boldsymbol{\theta}} \frac{\Delta\theta_i \, \Delta\theta_j}{1 + \mathrm{I}(i=j)} \right)
\end{aligned}
$$

The proof follows from the application of the bound to Taylor's expansion. $\qquad\square$

**Theorem 3.3.** *Let the update of the parameters be* $\boldsymbol{\theta}' = \boldsymbol{\theta} + \alpha\nabla_{\boldsymbol{\theta}}J_{\mu}(\boldsymbol{\theta})$. *Then for any stationary policy* $\pi(a|s,\boldsymbol{\theta})$ *and any starting state distribution* $\mu$, *the difference in performance between* $\pi_{\boldsymbol{\theta}}$ *and* $\pi_{\boldsymbol{\theta}'}$ *is lower bounded by:*

$$
J_{\mu}(\boldsymbol{\theta}') - J_{\mu}(\boldsymbol{\theta}) \geq \alpha\left\|\nabla_{\boldsymbol{\theta}}J_{\mu}(\boldsymbol{\theta})\right\|_2^2
$$

$$
+ \frac{\alpha^2}{1-\gamma}\int_{\mathcal{S}}d_{\mu}^{\pi_{\boldsymbol{\theta}}}(s)\int_{\mathcal{A}}\inf_{c\in(0,1)}\left(\sum_{i,j=1}^{m}\left.\frac{\partial^2\pi(a|s,\boldsymbol{\theta})}{\partial\theta_i\partial\theta_j}\right|_{\boldsymbol{\theta}+c\Delta\boldsymbol{\theta}}\frac{\Delta\theta_i\,\Delta\theta_j}{1+I(i=j)}\right)Q^{\pi_{\boldsymbol{\theta}}}(s,a)\mathrm{d}a\mathrm{d}s
$$

$$
- \frac{\gamma\left\|Q^{\pi_{\boldsymbol{\theta}}}\right\|_{\infty}}{2(1-\gamma)^2}\left(\alpha\sup_{s\in\mathcal{S}}\int_{\mathcal{A}}\left|\nabla_{\boldsymbol{\theta}}\pi(a|s,\boldsymbol{\theta})^{\scriptscriptstyle\mathsf{T}}\nabla_{\boldsymbol{\theta}}J_{\mu}(\boldsymbol{\theta})\right|\mathrm{d}a\right.
$$

$$
\left.+\alpha^2\sup_{s\in\mathcal{S}}\int_{\mathcal{A}}\left|\sup_{c\in(0,1)}\left(\sum_{i,j=1}^{m}\left.\frac{\partial^2\pi(a|s,\boldsymbol{\theta})}{\partial\theta_i\partial\theta_j}\right|_{\boldsymbol{\theta}+c\Delta\boldsymbol{\theta}}\frac{\Delta\theta_i\,\Delta\theta_j}{1+I(i=j)}\right)\right|\mathrm{d}a\right)^2.
$$

*Proof.* The first part of the proof is devoted to the derivation of the inequality reported in the theorem. Then, the positiveness and uniqueness of the optimal learning step is proved. Exploiting simple algebraic relationships and result in Lemma 3.2, Lemma 3.1 can be restated as:

$$
J_{\mu}(\boldsymbol{\theta}') - J_{\mu}(\boldsymbol{\theta}) \geq \frac{1}{1-\gamma}\int_{\mathcal{S}}d_{\mu}^{\pi}(s)\int_{\mathcal{A}}\left(\alpha\nabla_{\boldsymbol{\theta}}\pi(a|s,\boldsymbol{\theta})^{\scriptscriptstyle\mathsf{T}}\nabla_{\boldsymbol{\theta}}J_{\mu}(\boldsymbol{\theta}) + R_1(\Delta\boldsymbol{\theta})\right)Q^{\pi}(s,a)\mathrm{d}a\mathrm{d}s
$$

$$
- \frac{\gamma}{2(1-\gamma)^2}\left\|\pi_{\boldsymbol{\theta}'} - \pi_{\boldsymbol{\theta}}\right\|_{\infty}^2\left\|Q^{\pi_{\boldsymbol{\theta}}}\right\|_{\infty}
$$

$$
= \alpha\left\|\nabla_{\boldsymbol{\theta}}J_{\mu}(\boldsymbol{\theta})\right\|_2^2 + \frac{1}{1-\gamma}\int_{\mathcal{S}}d_{\mu}^{\pi}(s)\int_{\mathcal{A}}R_1(\Delta\boldsymbol{\theta})Q^{\pi}(s,a)\mathrm{d}a\mathrm{d}s
$$

$$
- \frac{\gamma}{2(1-\gamma)^2}\left\|\pi_{\boldsymbol{\theta}'} - \pi_{\boldsymbol{\theta}}\right\|_{\infty}^2\left\|Q^{\pi_{\boldsymbol{\theta}}}\right\|_{\infty},
$$

where last equality follows from the manipulation of the first–order approximation of the policy:

$$
\frac{\alpha}{1-\gamma}\int_{\mathcal{S}}d_{\mu}^{\pi}(s)\int_{\mathcal{A}}\nabla_{\boldsymbol{\theta}}\pi(a|s,\boldsymbol{\theta})^{\scriptscriptstyle\mathsf{T}}\nabla_{\boldsymbol{\theta}}J_{\mu}(\boldsymbol{\theta})\,Q^{\pi}(s,a)\mathrm{d}a\mathrm{d}s
$$

$$
= \frac{\alpha}{1-\gamma}\int_{\mathcal{S}}d_{\mu}^{\pi}(s)\int_{\mathcal{A}}\sum_{i=1}^{m}\frac{\partial\pi(a|s,\boldsymbol{\theta})}{\partial\theta_i}\frac{\partial J_{\mu}(\boldsymbol{\theta})}{\partial\theta_i}\,Q^{\pi}(s,a)\mathrm{d}a\mathrm{d}s
$$

$$
= \frac{\alpha}{1-\gamma}\sum_{i=1}^{m}\int_{\mathcal{S}}d_{\mu}^{\pi}(s)\int_{\mathcal{A}}\frac{\partial\pi(a|s,\boldsymbol{\theta})}{\partial\theta_i}\frac{\partial J_{\mu}(\boldsymbol{\theta})}{\partial\theta_i}\,Q^{\pi}(s,a)\mathrm{d}a\mathrm{d}s
$$

$$
= \frac{\alpha}{1-\gamma}\sum_{i=1}^{m}\left[\int_{\mathcal{S}}d_{\mu}^{\pi}(s)\int_{\mathcal{A}}\frac{\partial\pi(a|s,\boldsymbol{\theta})}{\partial\theta_i}\,Q^{\pi}(s,a)\mathrm{d}a\mathrm{d}s\right]\frac{\partial J_{\mu}(\boldsymbol{\theta})}{\partial\theta_i}
$$

$$
= \alpha\sum_{i=1}^{m}\left(\frac{\partial J_{\mu}(\boldsymbol{\theta})}{\partial\theta_i}\right)^2 = \alpha\left\|\nabla_{\boldsymbol{\theta}}J_{\mu}(\boldsymbol{\theta})\right\|_2^2
$$

To complete the derivation of the bound it is sufficient to notice that the difference between two policies can be upper bounded taking the maximum of the Lagrange remainder (as done in Lemma 3.2 for the lower bound):

$$
\left\|\pi_{\boldsymbol{\theta}'} - \pi_{\boldsymbol{\theta}}\right\|_{\infty} = \sup_{s\in\mathcal{S}}\int_{\mathcal{A}}\left|\pi(\cdot|s,\boldsymbol{\theta}') - \pi(\cdot|s,\boldsymbol{\theta})\right|\mathrm{d}a
$$

$$
\leq \alpha\sup_{s\in\mathcal{S}}\int_{\mathcal{A}}\left|\nabla_{\boldsymbol{\theta}}\pi(a|s,\boldsymbol{\theta})^{\scriptscriptstyle\mathsf{T}}\nabla_{\boldsymbol{\theta}}J_{\mu}(\boldsymbol{\theta})\right|\mathrm{d}a
$$

$$
+ \alpha^2\sup_{s\in\mathcal{S}}\int_{\mathcal{A}}\left|\sup_{c\in(0,1)}\left(\sum_{i,j=1}^{m}\left.\frac{\partial^2\pi(a|s,\boldsymbol{\theta})}{\partial\theta_i\partial\theta_j}\right|_{\boldsymbol{\theta}+c\Delta\boldsymbol{\theta}}\frac{\Delta\theta_i\,\Delta\theta_j}{1+\mathrm{I}(i=j)}\right)\right|\mathrm{d}a
$$

Finally:

$$J_\mu(\boldsymbol{\theta}') - J_\mu(\boldsymbol{\theta}) \geq \alpha \left\| \nabla_{\boldsymbol{\theta}} J_\mu(\boldsymbol{\theta}) \right\|_2^2$$

$$+ \frac{\alpha^2}{1-\gamma} \int_{\mathcal{S}} d_\mu^{\pi_{\boldsymbol{\theta}}}(s) \int_{\mathcal{A}} \inf_{c \in (0,1)} \left( \sum_{i,j=1}^{m} \frac{\partial^2 \pi(a|s,\boldsymbol{\theta})}{\partial \theta_i \partial \theta_j} \bigg|_{\boldsymbol{\theta}+c\Delta\boldsymbol{\theta}} \frac{\Delta\theta_i \, \Delta\theta_j}{1 + \mathrm{I}(i=j)} \right) Q^{\pi_{\boldsymbol{\theta}}}(s,a) \mathrm{d}a \mathrm{d}s$$

$$- \frac{\gamma \left\| Q^{\pi_{\boldsymbol{\theta}}} \right\|_\infty}{2(1-\gamma)^2} \left( \alpha \sup_{s \in \mathcal{S}} \int_{\mathcal{A}} \left| \nabla_{\boldsymbol{\theta}} \pi(a|s,\boldsymbol{\theta})^{\mathsf{T}} \nabla_{\boldsymbol{\theta}} J_\mu(\boldsymbol{\theta}) \right| \mathrm{d}a \right.$$

$$\left. + \alpha^2 \sup_{s \in \mathcal{S}} \int_{\mathcal{A}} \left| \sup_{c \in (0,1)} \left( \sum_{i,j=1}^{m} \frac{\partial^2 \pi(a|s,\boldsymbol{\theta})}{\partial \theta_i \partial \theta_j} \bigg|_{\boldsymbol{\theta}+c\Delta\boldsymbol{\theta}} \frac{\Delta\theta_i \, \Delta\theta_j}{1 + \mathrm{I}(i=j)} \right) \right| \mathrm{d}a \right)^2$$

$\square$

It is worth to show that exist only one positive value that maximize the previous bound. The bound stated in Theorem 3.3 is a fourth–order polynomial of step size $\alpha$ whose stationary points, i.e., the roots of a third–order polynomial $ax^3 + bx^2 + cx + d$, can be compute in closed form. If the product $a \cdot d$ is negative, the existence of at least one real positive solution is guaranteed. It is easy to evaluate the sign of the coefficient of degree zero, two and three of the third–order polynomial that are positive, negative and negative, respectively. Thus, the existence of a positive real solution is guaranteed. In order to demonstrate the uniqueness of the real positive solution it is possible to exploit the Descartes' rule of sign. According to that, given a polynomial with real coefficients order by descending degree, the number of positives roots equals the number of sign changes, or a value less than that by some multiple of 2. The sign of the coefficient $c$ cannot be determined a priori. However, for any value $c$, the number of sign changes is equal to 1 (recall that $a, b \leq 0$ and $d \geq 0$. Thus, the existence and uniqueness of the real positive root is proved.

## 2 The Gaussian Policy Model

**Lemma 4.1.** *For any Gaussian policy $\pi(a|s,\boldsymbol{\theta}) \sim N(\boldsymbol{\theta}^T\boldsymbol{\phi}(s), \sigma^2)$, the second order derivative of the policy can be bounded as follows:*

$$\left| \frac{\partial \pi(a|s,\boldsymbol{\theta})}{\partial \theta_i \partial \theta_j} \right| \leq \frac{|\phi_i(s)\phi_j(s)|}{\sqrt{2\pi}\sigma^3}, \quad \forall \boldsymbol{\theta} \in \mathbb{R}^m, \forall a \in \mathcal{A}.$$

*This result allows to restate Lemma 3.2 in the case of Gaussian policies:*

$$\pi(a|s,\boldsymbol{\theta}') - \pi(a|s,\boldsymbol{\theta}) \geq \alpha \nabla_{\boldsymbol{\theta}} \pi(a|s,\boldsymbol{\theta})^T \nabla_{\boldsymbol{\theta}} J_\mu(\boldsymbol{\theta}) - \frac{\alpha^2}{\sqrt{2\pi}\sigma^3} \left( |\nabla_{\boldsymbol{\theta}} J_\mu(\boldsymbol{\theta})|^T |\boldsymbol{\phi}(s)| \right)^2 .$$

*Proof.* The second order derivative of a Gaussian function is explicitly given by:

$$\frac{\partial^2 \pi(a|s,\boldsymbol{\theta})}{\partial \theta_i \partial \theta_j} = \frac{\pi(a|s,\boldsymbol{\theta})}{\sigma^2} \left( \frac{\left(a - \boldsymbol{\theta}^{\mathsf{T}}\boldsymbol{\phi}(s)\right)^2}{\sigma^2} - 1 \right) \phi_i(s)\phi_j(s), \tag{1}$$

It is easy to verify that the stationary point of the second–order derivative of a Gaussian distribution with mean $\mu$ and standard deviation $\sigma$ are $\mu = 0$ and $\mu = a \pm \sigma\sqrt{3}$. Plugging these results into Equation (1), we get:

$$-\frac{\phi_i(s)\phi_j(s)}{\sqrt{2\pi}\sigma^3}, \quad \frac{2e^{-\frac{3}{2}}\phi_i(s)\phi_j(s)}{\sqrt{2\pi}\sigma^3}.$$

The nature (maximum or minimum) of each point depends on the sign of the product $\phi_i(s)\phi_j(s)$ in each state $s$. As a consequence, we can state that the second order derivative is uniformly bounded by the maximum absolute value, i.e., $\frac{|\phi_i(s)\phi_j(s)|}{\sqrt{2\pi}\sigma^3}$.

The second part of the proof follows directly from the application of this result to Lemma 3.2, notice that $\left| \sum_i x_i \right| \leq \sum_i |x_i|$, for any $i$, $x_i$. $\square$

**Theorem 4.3.** *For any starting state distribution $\mu$, and any pair of stationary Gaussian policies $\pi_{\boldsymbol{\theta}} \sim N(\boldsymbol{\theta}^T\boldsymbol{\phi}(s), \sigma^2)$ and $\pi_{\boldsymbol{\theta}'} \sim N(\boldsymbol{\theta'}^T\boldsymbol{\phi}(s), \sigma^2)$, so that $\boldsymbol{\theta}' = \boldsymbol{\theta} + \alpha\nabla_{\boldsymbol{\theta}}J_\mu(\boldsymbol{\theta})$ and under Assumption 4.1, the difference between the performance of $\pi_{\boldsymbol{\theta}'}$ and $\pi_{\boldsymbol{\theta}}$ can be lower bounded by:*

$$J_\mu(\boldsymbol{\theta}') - J_\mu(\boldsymbol{\theta}) \geq \alpha \left\| \nabla_{\boldsymbol{\theta}}J_\mu(\boldsymbol{\theta}) \right\|_2^2$$
$$- \alpha^2 \left( \frac{1}{(1-\gamma)\sqrt{2\pi}\sigma^3} \int_{\mathcal{S}} d_\mu^{\pi_{\boldsymbol{\theta}}}(s) \left( |\nabla_{\boldsymbol{\theta}}J_\mu(\boldsymbol{\theta})|^T |\boldsymbol{\phi}(s)| \right)^2 \int_{\mathcal{A}} Q^{\pi_{\boldsymbol{\theta}}}(s,a)\mathrm{d}a\mathrm{d}s \right.$$
$$\left. + \frac{\gamma M_\phi^2}{2(1-\gamma)^2\sigma^2} \left\| \nabla_{\boldsymbol{\theta}}J_\mu(\boldsymbol{\theta}) \right\|_1^2 \left\| Q^{\pi_{\boldsymbol{\theta}}} \right\|_\infty \right).$$

*Proof.* For any state $s \in \mathcal{S}$, it is possible to use the Kullback–Liebler divergence

$$\mathrm{H}(P\|Q) = \int_X P(x) \log \frac{P(x)}{Q(x)}\mathrm{d}x,$$

to express the difference between policy $\pi(a|s, \boldsymbol{\theta}')$ and policy $\pi(a|s, \boldsymbol{\theta})$:

$$\mathrm{H}\big(\pi(\cdot|s, \boldsymbol{\theta}')\big\|\pi(\cdot|s, \boldsymbol{\theta})\big) = \frac{1}{2\sigma^2} \left( (\boldsymbol{\theta} + \alpha\nabla_{\boldsymbol{\theta}}J_\mu(\boldsymbol{\theta}))^T\boldsymbol{\phi}(s) - \boldsymbol{\theta}^T\boldsymbol{\phi}(s) \right)^2$$
$$= \frac{1}{2}\alpha^2 \left( \frac{\nabla_{\boldsymbol{\theta}}J_\mu(\boldsymbol{\theta})^T\boldsymbol{\phi}(s)}{\sigma} \right)^2 \quad \forall s \in \mathcal{S}.$$

In order to make explicit the dependence on $\alpha$ in Lemma 3.1 we need to manipulate the $L_\infty-$ norm between two policies. Recall that, for any distribution $P$ and $Q$ on an arbitrary set, the Pinsker's inequality (sometimes known as Pinsker–Csiszár–Kullback [1, 2, 3]) relates the Kullback–Liebler divergence $\mathrm{H}(P\|Q)$ and the variational divergence $V(P,Q) = \|P - Q\|_1$ by $\mathrm{H}(P\|Q) \geq \frac{1}{2}[V(P,Q)]^2$. Exploiting the Pinsker's inequality we can bound the $L_\infty-$norm between the current policy $\pi(a|s, \boldsymbol{\theta})$ and the improved policy $\pi(a|s, \boldsymbol{\theta}')$ as:

$$\|\pi_{\boldsymbol{\theta}'} - \pi_{\boldsymbol{\theta}}\|_\infty^2 = \sup_{s\in\mathcal{S}} \left\| \pi(\cdot|s, \boldsymbol{\theta}') - \pi(\cdot|s, \boldsymbol{\theta}) \right\|_1^2$$
$$\leq \sup_{s\in\mathcal{S}} \left( 2\mathrm{H}\big(\pi(\cdot|s, \boldsymbol{\theta}')\big\|\pi(\cdot|s, \boldsymbol{\theta})\big) \right)$$
$$= \frac{\alpha^2}{\sigma^2} \sup_{s\in\mathcal{S}} \left( \nabla_{\boldsymbol{\theta}}J_\mu(\boldsymbol{\theta})^T\boldsymbol{\phi}(s) \right)^2.$$

As a consequence of Assumption 4.1, we can state that:

$$\sup_{s\in\mathcal{S}} \left( \nabla_{\boldsymbol{\theta}}J_\mu(\boldsymbol{\theta})^T\boldsymbol{\phi}(s) \right)^2 = \sup_{s\in\mathcal{S}} \left( \sum_i \frac{\partial J_\mu(\boldsymbol{\theta})}{\partial\theta_i}\phi_i(s) \right)^2 \leq M_\phi^2 \|\nabla_{\boldsymbol{\theta}}J_\mu(\boldsymbol{\theta})\|_1^2.$$

The proof follows from the manipulation of Lemma 3.1 through the bound on the $L_\infty-$norm and the result in Lemma 4.1. $\qquad\square$

**Corollary 4.4.** *The performance lower bound provided in Theorem 4.3 is maximized by choosing the following step size:*

$$\alpha^* = \frac{(1-\gamma)^2\sqrt{2\pi}\sigma^3 \|\nabla_{\boldsymbol{\theta}}J_\mu(\boldsymbol{\theta})\|_2^2}{\gamma\sqrt{2\pi}\sigma M_\phi^2 \|\nabla_{\boldsymbol{\theta}}J_\mu(\boldsymbol{\theta})\|_1^2 \|Q^{\pi_{\boldsymbol{\theta}}}\|_\infty + 2(1-\gamma)\int_{\mathcal{S}} d_\mu^{\pi_{\boldsymbol{\theta}}}(s) \left( |\nabla_{\boldsymbol{\theta}}J_\mu(\boldsymbol{\theta})|^T |\boldsymbol{\phi}(s)| \right)^2 \int_{\mathcal{A}} Q^{\pi_{\boldsymbol{\theta}}}(s,a)\mathrm{d}a\mathrm{d}s},$$

*that guarantees the following policy performance improvement*

$$J_\mu(\boldsymbol{\theta}') - J_\mu(\boldsymbol{\theta}) \geq \frac{1}{2}\alpha^* \|\nabla_{\boldsymbol{\theta}}J_\mu(\boldsymbol{\theta})\|_2^2.$$

*Proof.* Consider the bound stated in Theorem 3.3. The term $\alpha^*$ is the value of $\alpha$ that maximizes this bound, i.e., the value that sets the partial derivative w.r.t. $\alpha$ to zero:

$$\frac{\partial B}{\partial \alpha} = \|\nabla_{\boldsymbol{\theta}} J_\mu(\boldsymbol{\theta})\|_2^2$$

$$- 2\alpha \left( \frac{1}{(1-\gamma)\sqrt{2\pi}\sigma^3} \int_{\mathcal{S}} d_\mu^{\pi_{\boldsymbol{\theta}}}(s) \left( |\nabla_{\boldsymbol{\theta}} J_\mu(\boldsymbol{\theta})|^{\top} |\boldsymbol{\phi}(s)| \right)^2 \int_{\mathcal{A}} Q^{\pi_{\boldsymbol{\theta}}}(s,a) \mathrm{d}a \mathrm{d}s \right.$$

$$\left. + \frac{\gamma M_\phi^2}{2(1-\gamma)^2 \sigma^2} \|\nabla_{\boldsymbol{\theta}} J_\mu(\boldsymbol{\theta})\|_1^2 \|Q^{\pi_{\boldsymbol{\theta}}}\|_\infty \right).$$

that leads to:

$$\alpha^* = \frac{(1-\gamma)^2 \sqrt{2\pi}\sigma^3 \|\nabla_{\boldsymbol{\theta}} J_\mu(\boldsymbol{\theta})\|_2^2}{\gamma\sqrt{2\pi}\sigma M_\phi^2 \|\nabla_{\boldsymbol{\theta}} J_\mu(\boldsymbol{\theta})\|_1^2 \|Q^{\pi_{\boldsymbol{\theta}}}\|_\infty + 2(1-\gamma) \int_{\mathcal{S}} d_\mu^{\pi_{\boldsymbol{\theta}}}(s) \left( |\nabla_{\boldsymbol{\theta}} J_\mu(\boldsymbol{\theta})|^{\top} |\boldsymbol{\phi}(s)| \right)^2 \int_{\mathcal{A}} Q^{\pi_{\boldsymbol{\theta}}}(s,a) \mathrm{d}a \mathrm{d}s}$$

By replacing the value $\alpha^*$ in the bound, we can derive the guaranteed policy improvement:

$$J_\mu(\boldsymbol{\theta}') - J_\mu(\boldsymbol{\theta}) \geq \frac{1}{2}\alpha^* \|\nabla_{\boldsymbol{\theta}} J_\mu(\boldsymbol{\theta})\|_2^2.$$

$\square$

# 3 Approximate Framework

**Corollary 5.1.** *For any starting state distribution $\mu$, and any pair of stationary Gaussian policies $\pi_{\boldsymbol{\theta}} \sim \mathcal{N}(\boldsymbol{\theta}^T \boldsymbol{\phi}(s), \sigma^2)$ and $\pi_{\boldsymbol{\theta}'} \sim \mathcal{N}(\boldsymbol{\theta}'^T \boldsymbol{\phi}(s), \sigma^2)$, so that $\boldsymbol{\theta}' = \boldsymbol{\theta} + \alpha \nabla_{\boldsymbol{\theta}} J_\mu(\boldsymbol{\theta})$ and under Assumption 4.1, the difference between the performance of $\pi_{\boldsymbol{\theta}'}$ and $\pi_{\boldsymbol{\theta}}$ can be lower bounded by:*

$$J_\mu(\boldsymbol{\theta}') - J_\mu(\boldsymbol{\theta}) \geq \alpha \|\nabla_{\boldsymbol{\theta}} J_\mu(\boldsymbol{\theta})\|_2^2 - \alpha^2 \frac{R M_\phi^2 \|\nabla_{\boldsymbol{\theta}} J_\mu(\boldsymbol{\theta})\|_1^2}{(1-\gamma)^2 \sigma^2} \left( \frac{|\mathcal{A}|}{\sqrt{2\pi}\sigma} + \frac{\gamma}{2(1-\gamma)} \right),$$

*that is maximized by the following step size value:*

$$\widehat{\alpha}^* = \frac{(1-\gamma)^3 \sqrt{2\pi}\sigma^3 \|\nabla_{\boldsymbol{\theta}} J_\mu(\boldsymbol{\theta})\|_2^2}{\left( \gamma\sqrt{2\pi}\sigma + 2(1-\gamma)|\mathcal{A}| \right) R M_\phi^2 \|\nabla_{\boldsymbol{\theta}} J_\mu(\boldsymbol{\theta})\|_1^2}.$$

*Proof.* Under the assumption of positive reward, for every state $s \in \mathcal{S}$ and every action $a \in \mathcal{A}$, the $Q$–function belongs to $\left[ 0, \frac{R}{1-\gamma} \right]$. As a consequence, the integral of the $Q$–function over the action space and the $L_\infty$–norm of the $Q$–function are upper bounded by $\frac{|\mathcal{A}|R}{1-\gamma}$ and $\frac{R}{1-\gamma}$, respectively. Furthermore, exploiting Assumption 4.1, we can restate the policy performance improvement as:

$$J_\mu(\boldsymbol{\theta}') - J_\mu(\boldsymbol{\theta}) \geq \alpha \|\nabla_{\boldsymbol{\theta}} J_\mu(\boldsymbol{\theta})\|_2^2$$

$$- \alpha^2 \left( \frac{|\mathcal{A}|\,R}{(1-\gamma)^2 \sqrt{2\pi}\sigma^3} \int_{\mathcal{S}} d_\mu^{\pi_{\boldsymbol{\theta}}}(s) \left( |\nabla_{\boldsymbol{\theta}} J_\mu(\boldsymbol{\theta})|^{\top} |\boldsymbol{\phi}(s)| \right)^2 \mathrm{d}s \right.$$

$$\left. + \frac{\gamma R M_\phi^2}{2(1-\gamma)^3 \sigma^2} \|\nabla_{\boldsymbol{\theta}} J_\mu(\boldsymbol{\theta})\|_1^2 \right) \tag{2}$$

$$\geq \alpha \|\nabla_{\boldsymbol{\theta}} J_\mu(\boldsymbol{\theta})\|_2^2 - \alpha^2 \frac{R M_\phi^2 \|\nabla_{\boldsymbol{\theta}} J_\mu(\boldsymbol{\theta})\|_1^2}{(1-\gamma)^2 \sigma^2} \left( \frac{|\mathcal{A}|}{\sqrt{2\pi}\sigma} + \frac{\gamma}{2(1-\gamma)} \right).$$

The new optimal learning step is derived by setting the partial derivative w.r.t. $\alpha$ of previous inequality to zero:

$$\widehat{\alpha}^* = \frac{(1-\gamma)^3 \sqrt{2\pi}\sigma^3 \|\nabla_{\boldsymbol{\theta}} J_\mu(\boldsymbol{\theta})\|_2^2}{\left( \gamma\sqrt{2\pi}\sigma + 2(1-\gamma)|\mathcal{A}| \right) R M_\phi^2 \|\nabla_{\boldsymbol{\theta}} J_\mu(\boldsymbol{\theta})\|_1^2}$$

It is easy to observe that the guaranteed policy performance is again at least $\frac{1}{2}\widehat{\alpha}^* \|\nabla_{\boldsymbol{\theta}} J_\mu(\boldsymbol{\theta})\|_2^2$.

$\square$

**Theorem 5.2.** *Under the same assumptions of Corollary 5.1, and provided that it is available a policy gradient estimate $\widehat{\nabla}_{\boldsymbol{\theta}} J_\mu(\boldsymbol{\theta})$, so that $\mathbb{P}\left(\left|\nabla_{\theta_i} J_\mu(\boldsymbol{\theta}) - \widehat{\nabla}_{\theta_i} J_\mu(\boldsymbol{\theta})\right| \geq \epsilon_i\right) \leq \delta$, the difference between the performance of $\pi_{\boldsymbol{\theta}'}$ and $\pi_{\boldsymbol{\theta}}$ can be lower bounded at least with probability $(1-\delta)^m$:*

$$J_\mu(\boldsymbol{\theta}') - J_\mu(\boldsymbol{\theta}) \geq \alpha \left\|\widehat{\underline{\nabla_{\boldsymbol{\theta}} J_\mu}}(\boldsymbol{\theta})\right\|_2^2 - \alpha^2 \frac{RM_\phi^2 \left\|\widehat{\overline{\nabla_{\boldsymbol{\theta}} J_\mu}}(\boldsymbol{\theta})\right\|_1^2}{(1-\gamma)^2 \sigma^2} \left(\frac{|\mathcal{A}|}{\sqrt{2\pi}\sigma} + \frac{\gamma}{2(1-\gamma)}\right),$$

*that is maximized by the following step size value:*

$$\widehat{\alpha}^* = \frac{(1-\gamma)^3 \sqrt{2\pi}\sigma^3 \left\|\widehat{\underline{\nabla_{\boldsymbol{\theta}} J_\mu}}(\boldsymbol{\theta})\right\|_2^2}{\left(\gamma\sqrt{2\pi}\sigma + 2(1-\gamma)|\mathcal{A}|\right) RM_\phi^2 \left\|\widehat{\overline{\nabla_{\boldsymbol{\theta}} J_\mu}}(\boldsymbol{\theta})\right\|_1^2}.$$

*Proof.* We have access to an $\epsilon$–accurate estimation of the gradient. In order to preserve the sign of the inequality in Corollary 5.1 and take into account the approximation error, we need to decrease the L2–norm of the gradient and increment the L1–norm in the penalization term. If we treat the two terms in a separate way, we cannot do worse than suppose to have an over estimate of the positive term and an under estimate of the penalization term (both of an amount $\epsilon$). Under this worst case scenario, the correction term of each gradient component $\nabla_{\theta_i} J_\mu(\boldsymbol{\theta})$ is $-\epsilon_i$ and $\epsilon_i$ for the over and under estimate case, respectively. Then,

$$\|\nabla_{\boldsymbol{\theta}} J_\mu(\boldsymbol{\theta})\|_2^2 \geq \sum_i \left(\max\left(|\nabla_{\theta_i} J_\mu(\boldsymbol{\theta})| - \epsilon_i, 0\right)\right)^2 = \left\|\widehat{\underline{\nabla_{\boldsymbol{\theta}} J_\mu}}(\boldsymbol{\theta})\right\|_2^2$$

$$\|\nabla_{\boldsymbol{\theta}} J_\mu(\boldsymbol{\theta})\|_1^2 \leq \sum_i |\nabla_{\theta_i} J_\mu(\boldsymbol{\theta})| + \epsilon_i = \left\|\widehat{\overline{\nabla_{\boldsymbol{\theta}} J_\mu}}(\boldsymbol{\theta})\right\|_1^2$$

Notice that a saturation to $0$ is necessary in order to preserve the correctness of the inequality. The new bound on the policy performance improvement is obtained by substituting the corrected gradients in place of the original ones in Theorem 5.1. Such bound holds at least with probability $(1-\delta)^m$ that is the probability, assuming independent events, that all the approximation errors of the different gradient components are smaller than their respective $\epsilon$ value. If some correlation exists in the approximation errors, the actual probability will be higher. The optimal learning step is computed by maximizing the new bound. $\square$

**Lemma 5.3.** *[Adapted from Theorem 2 in [4]] Given a Gaussian policy $\pi(a|s, \boldsymbol{\theta}) \sim \mathcal{N}\left(\boldsymbol{\theta}^\top \phi(s), \sigma^2\right)$, under the assumption of uniformly bounded rewards and basis functions (Assumption 4.1), we have the following upper bound to the variance of the $i$–th component of the episodic REINFORCE gradient estimate $\widehat{\nabla}_{\theta_i} J_\mu^{RF}(\boldsymbol{\theta})$:*

$$Var\left(\widehat{\nabla}_{\theta_i} J_\mu^{RF}(\boldsymbol{\theta})\right) \leq \frac{R^2 M_\phi^2 H \left(1 - \gamma^H\right)^2}{N\sigma^2 (1-\gamma)^2}.$$

*Proof.* The proof follows from the result in [4] by notice that for any time $t$, $\|\phi(s_t)\|_2^2 \leq mM_\phi^2$ w.p. 1. $\square$

**Theorem 5.4.** *Given a Gaussian policy $\pi(a|s, \boldsymbol{\theta}) \sim \mathcal{N}\left(\boldsymbol{\theta}^\top \phi(s), \sigma^2\right)$, under the assumption of uniformly bounded rewards and basis functions (Assumption 4.1), using the following number of $H$–step trajectories:*

$$N = \frac{R^2 M_\phi^2 H \left(1 - \gamma^H\right)^2}{\delta\epsilon_i^2 \sigma^2 (1-\gamma)^2},$$

*the gradient estimate $\widehat{\nabla}_{\theta_i} J_\mu^{RF}(\boldsymbol{\theta})$ generated by REINFORCE is such that with probability $1 - \delta$:*

$$\left|\widehat{\nabla}_{\theta_i} J_\mu^{RF}(\boldsymbol{\theta}) - \nabla_{\theta_i} J_\mu(\boldsymbol{\theta})\right| \leq \epsilon_i.$$

*Proof.* Chebyshev's inequality implies that

$$\mathbb{P}\left(\left|\widehat{\nabla}_{\theta_i} J_\mu^{RF}(\boldsymbol{\theta}) - \nabla_{\theta_i} J_\mu(\boldsymbol{\theta})\right| \geq \epsilon_i\right) \leq \frac{\sigma^2}{\epsilon_i^2} = \frac{R^2 M_\phi^2 H \left(1 - \gamma^H\right)^2}{\epsilon^2 N \sigma^2 \left(1 - \gamma\right)^2} = \delta.$$

Solving the equation for $N$, we obtain

$$N = \frac{R^2 M_\phi^2 H \left(1 - \gamma^H\right)^2}{\delta \epsilon_i^2 \sigma^2 \left(1 - \gamma\right)^2}.$$

Hence, with probability of $1 - \delta$, the maximum deviation of the estimation of the $i$–th component of the gradient from the true mean is $\epsilon_i$. $\qquad\square$

**Lemma 5.5.** *Given a Gaussian policy $\pi(a|s, \boldsymbol{\theta}) \sim \mathcal{N}\left(\boldsymbol{\theta}^T \boldsymbol{\phi}(s), \sigma^2\right)$, under the assumption of uniformly bounded rewards and basis functions (Assumption 4.1), we have the following upper bound to the variance of the $i$–th component of the PGT gradient estimate $\widehat{\nabla}_{\theta_i} J_\mu^{PGT}(\boldsymbol{\theta})$:*

$$Var\left(\widehat{\nabla}_{\theta_i} J_\mu^{PGT}(\boldsymbol{\theta})\right) \leq \frac{R^2 M_\phi^2}{N \left(1 - \gamma\right)^2 \sigma^2} \left[\frac{1 - \gamma^{2H}}{1 - \gamma^2} + H\gamma^{2H} - 2\gamma^H \frac{1 - \gamma^H}{1 - \gamma}\right].$$

*Proof.* Let $f(t) = \nabla_{\boldsymbol{\theta}} \log \pi\left(a_t; s_t, \boldsymbol{\theta}\right)$, if the policy is Gaussian the $i$–th component of $f(\cdot)$ at time $t$ is given by:

$$f_i(t) = \nabla_{\theta_i} \log \pi\left(a_t; s_t, \boldsymbol{\theta}\right) = \frac{a - \boldsymbol{\theta}^T \boldsymbol{\phi}(s_t)}{\sigma^2} \phi_i(s_t).$$

Before to focus on the derivation of an upper bound to the variance of the $i$–th component, we need to introduce the term $\mathbb{T}$ that denotes the space of all the trajectories of length $H$ generated from the system. Since

$$Var\left(\widehat{\nabla}_{\theta_i} J_\mu^{PGT}(\boldsymbol{\theta})\right) = \frac{1}{N} Var\left(\sum_{t=1}^H f_i(t) \sum_{l=t}^H \gamma^{l-1} r_l\right) \tag{3}$$

we can just focus on the derivation of the variance of the $i$–th element for a single trajectory:

$$Var\left(\sum_{t=1}^H f_i(t) \sum_{l=t}^H \gamma^{l-1} r_l\right) \leq \mathbb{E}_{\mathbb{T}}\left[\left(\sum_{t=1}^H f_i(t) \sum_{l=t}^H \gamma^{l-1} r_l\right)^2\right]$$

$$= R^2 \mathbb{E}_{\mathbb{T}}\left[\left(\sum_{t=1}^H f_i(t) \left(\sum_{l=1}^H \gamma^{l-1} - \sum_{l=1}^{t-1} \gamma^{l-1}\right)\right)^2\right] = \frac{R^2}{(1 - \gamma)^2} \mathbb{E}_{\mathbb{T}}\left[\left(\sum_{t=1}^H f_i(t) \left(\gamma^{t-1} - \gamma^H\right)\right)^2\right]$$

$$= \frac{R^2}{(1 - \gamma)^2} \mathbb{E}_{\mathbb{T}}\left[\left(\sum_{t=1}^H \gamma^{t-1} f_i(t)\right)^2 + \gamma^{2H}\left(\sum_{t=1}^H f_i(t)\right)^2 - 2\gamma^H \sum_{t=1}^H \gamma^{t-1} f_i(t) \sum_{t=1}^H f_i(t)\right] \tag{4}$$

Let $\eta_{i,t} = \frac{a_t - \boldsymbol{\theta}^T \boldsymbol{\phi}(s_t)}{\sigma}$ for $t = 1, \ldots, H$. Note that $\eta_{i,1}, \ldots, \eta_{i,H}$ are independent standard normal variables. Moreover, given the entire history of basis functions $\{\phi_i(s_t)\}_{t=1}^H$, $\eta_{i,1}\phi_i(s_1), \ldots, \eta_{i,H}\phi_i(s_H)$ are independent normal variables with zero mean, i.e., $\mathbb{E}\left[\eta_{i,t}\phi_i(s_t)\right] = 0$. Then, exploiting the relationship $\eta_{i,t} = \frac{\sigma}{\phi_i(s_t)} \cdot f_i(t)$, we can state that:

$$\mathbb{E}_{\mathbb{T}}\left[\left(\sum_{t=1}^H \gamma^{t-1} f_i(t)\right)^2\right] = \mathbb{E}_{\mathbb{T}}\left[\sum_{t=1}^H \sum_{t'=1}^H \gamma^{t-1} \gamma^{t'-1} f_i(t) f_i(t')\right]$$

$$= \frac{1}{\sigma^2} \mathbb{E}_{\mathbb{T}}\left[\sum_{t=1}^H \sum_{t'=1}^H \gamma^{t-1} \gamma^{t'-1} \eta_{i,t} \eta_{i,t'} \phi_i(s_t) \phi_i(s_{t'})\right]$$

$$= \frac{1}{\sigma^2} \sum_{t=1}^H \mathbb{E}_{\mathbb{T}}\left[\gamma^{2(t-1)} \eta_{i,t}^2 \phi_i(s_t)^2\right] + \frac{1}{\sigma^2} \sum_{t=1}^H \sum_{t'=1; t' \neq t}^H \gamma^{t-1} \gamma^{t'-1} \mathbb{E}\left[\eta_{i,t}\phi_i(s_t)\right] \mathbb{E}\left[\eta_{i,t'}\phi_i(s_{t'})\right]$$

$$= \frac{1}{\sigma^2} \sum_{t=1}^H \gamma^{2(t-1)} \phi_i(s_t)^2$$

Note that last equality follows from the consideration that $\eta_{i,t}{}^2 \sim \chi^2(1)$ and $\mathbb{E}\left[\eta_{i,t}{}^2\right] = 1$. Performing the same consideration for all the terms in equation (4) leads to:

$$Var\left(\sum_{t=1}^{H} f_i(t) \sum_{l=t}^{H} \gamma^{l-1} r_l\right)$$

$$= \frac{R^2}{(1-\gamma)^2}\left[\frac{1}{\sigma^2}\sum_{t=1}^{H}\gamma^{2(t-1)}\phi_i(s_t)^2 + \frac{\gamma^{2H}}{\sigma^2}\sum_{t=1}^{H}\phi_i(s_t)^2 - \frac{2\gamma^H}{\sigma^2}\sum_{t=1}^{H}\gamma^{t-1}\phi_i(s_t)^2\right]$$

$$\leq \frac{R^2 M_\phi^2}{(1-\gamma)^2\sigma^2}\left[\frac{1-\gamma^{2H}}{1-\gamma^2} + H\gamma^{2H} - 2\gamma^H\frac{1-\gamma^H}{1-\gamma}\right]$$

where last inequality exploits the assumption of uniformly bounded basis functions. The proof follows from the substitution of last result in Equation 3. $\qquad\square$

**Theorem 5.6.** *Given a Gaussian policy $\pi(a|s,\boldsymbol{\theta}) \sim \mathcal{N}\left(\boldsymbol{\theta}^T\boldsymbol{\phi}(s),\sigma^2\right)$, under the assumption of uniformly bounded rewards and basis functions (Assumption 4.1), using the following number of $H$–step trajectories:*

$$N = \frac{R^2 M_\phi^2}{\delta\epsilon_i^2\sigma^2(1-\gamma)^2}\left[\frac{1-\gamma^{2H}}{1-\gamma^2} + H\gamma^{2H} - 2\gamma^H\frac{1-\gamma^H}{1-\gamma}\right]$$

*the gradient estimate $\widehat{\nabla}_{\theta_i} J_\mu^{PGT}(\boldsymbol{\theta})$ generated by PGT is such that with probability $1-\delta$:*

$$\left|\widehat{\nabla}_{\theta_i} J_\mu^{PGT}(\boldsymbol{\theta}) - \nabla_{\theta_i} J_\mu(\boldsymbol{\theta})\right| \leq \epsilon_i.$$

*Proof.* The proof follows the same approach used to prove Theorem 5.4. $\qquad\square$

## 4 Numerical Simulations

In this section we show results related to some numerical simulations of policy gradient in the linear–quadratic Gaussian regulation (LQG) problem as formulated in [5]. The LQG problem is characterized by a transition model $s_{t+1} \sim \mathcal{N}\left(As_t + Ba_t, \sigma^2\right)$, Gaussian policy $a_t \sim \mathcal{N}\left(\theta \cdot s, \sigma^2\right)$ and quadratic reward $r_t = -Qs_t^2 - Ra_t^2$. In our settings we put $A = B = 1$ and $Q = R = 1/2$. The range of state and action spaces is bounded to the interval $[-2, 2]$ and the initial state is drawn uniformly from the same range. This scenario is particularly instructive because it allows to exactly compute all the terms involved in the bounds. For this reason, we first present results in the exact scenario and then we move toward the approximated one.

Figure 1 and 2 report the trend of the learning step and the performance for each iteration, respectively. The scenario is the same exploited for the generation of Table 1 stated in the main article. In particular, we have reported the behavior of the exact gradient with different learning step for $\sigma = 1.75$. Configurations that have led to divergence are not depicted in the figures. It is worth to notice that the proposed auto–tuning approach is able to increment the value of the learning step in order to compensate the decrements of the gradient. Figure 2 shows that, when the learning step is tuned using the descendant rule $\alpha_t = \frac{\alpha_0}{t}$, the policy gradient is able to learn rapidly in the first iteration but the learning step becomes soon very small leading to a "stationary" situation in which no significant improvement to the performance is achieved.