[Reviews · NeurIPS 2013]

Submitted by Assigned_Reviewer_5

This paper provides a theoretically grounded adaptive step size for policy gradient methods in which the lower bound of the expected policy improvement is maximized. Approximated version of step size can be computed only by estimated gradient and thus the computational complexity would be same as ordinary policy gradient methods. There is a mild assumption that the standard deviation of Gaussian policy is fixed for deriving the lower bound. Experimental results show that the proposed method works well when the fixed SD is large enough.

Quality, clarity and originality:
The motivation of the paper is clearly described and its organization is reasonable. The paper provides potentially useful theoretical analyses such as Theorem 4.3 and Corollary 4.4 although I cannot check the derivation of lemmas and theorems in detail. The originality is definitely high.

Significance:
Newton method would be a standard approach for adaptive step-size but the second-order derivative of expected return with Gaussian policy is not guaranteed to be semi-positive definite. Thus, adaptive step-size based on the maximization of lower-bound would be a good alternative of ordinary policy gradient methods.
However, as the author(s) review in the introduction, there are several extensions of policy gradients for stable policy update, such as EM policy search. Thus, it would be much more convincing if there are more experimental comparisons with these methods.

Small comments and questions:
* The section of conclusion should be added.
* Is it really possible to converge to the optimal policy with only one update for \sigma=5 (in Table 1)?
Summary: The paper provides potentially useful theoretical analyses and good alternative of policy gradient methods. It would be much more convincing if there are more experimental comparisons with state-of-art methods such as EM policy search.

Submitted by Assigned_Reviewer_7

This paper investigates how to automatically set the step size of policy gradient methods, in particular focusing on the Gaussian policy case. The paper provides a lower bound on the difference in performance when taking a gradient step, and optimizes this lower bound with respect to the step size.

Overall, I like the approach of thoroughly studying the interplay between step size and policy performance. While the paper could be improved with clearer, more direct results, I think this is a fine, albeit incremental result for NIPS.

Quality. The paper contains a hefty amount of theoretical work, but sometimes lacks a certain focus, e.g. Theorem 3.3 seems extraneous to the main result. I think the paper would be improved with a clear directing line throughout and less "beating around the bush".

The experimental results are somewhat disappointing. While a meaningful comparison, they don't strongly confirm the superiority of the approach.

Clarity. The paper is sometimes hard to follow because some of the steps are relegated to the supplemental material or ignored altogether. For example, the authors argue repeatedly using stationary points without providing formal details.

Originality. This is original work.

Significance. I appreciate the idea of improving policy gradient methods by refining the step size, and doubly so by obtaining a theoretical grasp of the problem. But I think the results presented here are marginally significant.

Suggestion. The algorithmic aspect of this work (Section 5) comes in late, and seems mostly an afterthought. I think the paper could be much improved by emphasizing the algorithm, and subsequently deriving the quantities needed in the bound.
Summary: Overall, I like the approach of thoroughly studying the interplay between step size and policy performance. While the paper could be improved with clearer, more direct results, I think this is a fine, albeit incremental result for NIPS.

Submitted by Assigned_Reviewer_8

Overview
========
The paper proposes a lower bound on the expected performance gain for policy gradient methods. This lower bound can then used to optimize the step-size. The bound is given in general, specialized for Gaussian policies, and finally a version for REINFORCE and G(PO)MDP/PGT is given, which is also evaluated empirically on a simple LQG problem.

Quality
=======
The theoretical part is very nice with a lot of material. The experiment nicely illustrates the approach. It would be highly interesting to evaluate how well the adaptive step-size performs in motor control and robotic applications (but that is clearly out of the scope of this paper).

Clarity
=======
The paper reads well. The paper itself nicely manages to convey a notion of where the theorems and lemmas come from/how they are proven. I like the structure of the paper that goes from more theoretical to more and more applied. However, towards the end (Sect. 6) I got the feeling that it gets a bit crammed...

Originality
===========
To my knowledge nobody has looked into adaptive step-sizes for policy gradient methods from a theoretical point of view yet.

Significance
============
Even though tighter bounds would be nice (as mentioned in Sect. 6), the results are interesting from a more theoretical point of view and even show promise in experiments.

Minor Comments:
===============
Table 1: maybe you could add another line with the optimal step-size (determined by line search as discussed in Sect. 1.)

l 75: "do not need to tune a free parameter" is a bit misleading. It is true that this type of algorithms does not have a step-size but still e.g. the exploration magnitude needs tuning...

Table 1: maybe you could mark the "winners"

Table 1: 1 step for alpha* with sigma=5. is hard to believe/a very lucky coincidence

Table 2: How do you count the iterations here? The number of update steps or (number of update steps)*(number of trajectories)?

Table 2: maybe use e notation instead of >100,000

Table 2: especially here a comparison to the heuristics would have been interesting

Supplementary Material Fig. 1&2: maybe you could use a logarithmic iteration axis (such that we can see better what happens at the beginning with alpha=1e-5/t). Axis labels would also be nice.

Public Review:
==============
The author response addresses most of the above points. The notes on spelling and grammar mistakes as well as minor formatting/LaTeX suggestions were removed.
Summary: The paper tackles a very interesting unsolved problem from a theoretical point of view and manges to get results that have great potential for practical applications.

Submitted by Assigned_Reviewer_9

This paper proposes a new policy-gradient reinforcement learning method in which the step size is set adaptively by maximizing a lower bound on the expected performance gain. The authors first consider the case where the policy gradient is known and establish a lower bound on the difference in performance between the original policy and the modified policy resulting from an update that follows the policy gradient. Since the above bound is a fourth-order polynomial, the authors then consider the special case in which the stochastic policy is represented as a Gaussian distribution with a fixed standard deviation and a mean that is a linear combination of the state features. In this case, they derive a lower bound that is quadratic in the step size and has a single maximum for positive step sizes. Next, they remove the assumption that the policy gradient is known and establish a high-probability lower bound given an epsilon-optimal estimate of the policy gradient. Finally, they show how two existing techniques, REINFORCE and G{PO)MDP/PGT, can provide such policy gradient estimates, leading to an adaptive step size policy gradient method that does not require a model of the environment and can learn only from sample trajectories. The proposed method is evaluated in a simple simulation experiment using a 1-dimensional toy problem.

Overall, this is a high-quality, well written paper. The authors make a good argument that most research on policy-gradient methods has focused on finding better gradient estimators and thus there remains room for improvement in automating the step size in such algorithms. The theoretical results in the paper make a substantial contribution by providing a principled framework for the automatic selection of such step sizes by maximizing lower bounds on expected performance gain. The paper contributes not only a specific algorithm but more general results that could be used to derive algorithms with different policy representations and gradient estimators.

The main weakness in the paper is the empirical evaluation, which considers only a toy problem involving the optimization of one policy parameter. Results obtained using the true gradient show that manual choices of the step size can lead to slow learning (too small) or divergence (too big) whereas the adaptive method always converges within the threshold. For large values of the standard deviation of the Gaussian policy, the adaptive method performs better than any of the fixed step-sizes tested. However, for small standard deviations it performs worse, which the authors suggest reflects an inherent trade-off between the determinism of the policy and tightness of the lower bound.

The authors also present results for the case where the true gradient is not known and the REINFORCE and PGT estimators are used instead. However, it is not clear what these results demonstrate, other than that, unsurprisingly, increasing the number of trajectories used to estimate the gradient improves the gradient estimate and thus learning performance. More worryingly, however, the results also show that even with many trajectories, the errors are quite large, leading to loose bounds that prevent strong performance. This casts doubt on whether the results have practical implications or are only of theoretical interest.

Furthermore, the empirical results compare only to fixed-step size variants and not to other baseline methods. In particular, the EM-based policy-gradient-like methods of Kobers & Peters and Vlassis et al. mentioned in the introduction seem like particularly relevant baselines since they do not need to tune a free parameter.





Summary: Well written paper with a substantial theoretical contribution. The empirical evaluation is preliminary and yields mixed results for the proposed method.
Author Feedback

Author rebuttal: We thank the reviewers for their insightful comments and valuable suggestions to improve our paper.
Below we first provide clarifications to general doubts that are common to some reviews and then we answer to each reviewer's specific questions.

We are aware that our paper provides many contributions (theoretical, algorithmic, and empirical) that needed to be condensed to meet the constraints of the conference format.
Actually, we tried to focus the paper as much as possible by cutting other contributions that we have included in the first drafts of the paper (e.g., the case where the Gaussian standard deviation is not fixed and the derivation for the Gibbs policy model).
Furthermore, to save some space, we decided to merge the empirical results and conclusions sections.
We agree that this final section is quite compressed, but, in our opinion, the alternative would have been to drop the experimental section. We thought that showing some empirical evaluation of the proposed approach is a valuable contribution even in a mainly theoretical paper.

We agree with the reviewers that adding some comparisons with other related approaches would improve the significance of the empirical evaluation.
In particular, we will add in the final version of the paper a comparison with EM policy search.
For what concerns a comparison with natural policy gradient (NPG), the problem is that NPG itself relies on the choice of the step size.
What we could do is to extend our approach to NPG, but this would be another contribution that cannot fit in this eight-page paper, but could be considered for future works.

The surprising one-step convergence happening with sigma equal to 5 is not a mistake.
It is worth to notice that we stop the algorithm when the accuracy is below the 0.01 threshold w.r.t. the optimal solution.
Clearly, even with suitable fixed step sizes it is possible to achieve such a behavior (e.g., when sigma=5 by taking a fixed step size in the interval [3.28e-6,3.39e-6] attains a 1-step convergence), but small perturbations may lead to divergence. Our adaptive approach does not require to hand-tune the step size thus leading to much more robust policy gradient methods.

Assigned_Reviewer_5

We think that the main issues raised by the reviewer have been clarified by the arguments reported above.


Assigned_Reviewer_7

The motivation for the theoretical results introduced in Section 3 is twofold: from one side we introduce general results that can provide the basis for future works (e.g., using the theorem 3.3 and specialize it for other policy models), while from the other side it introduces concepts that simplify the presentation of the following sections.

The goal of the empirical evaluation was not to prove the superiority of our proposal, but to verify that an automatic step-size selection can add robustness to policy gradient methods.
In fact, if the problem is well-known or many simulations can be performed in a short time, the step-size parameter can finely tuned by hand.
Our proposal provides an automatic way to set the step size, obtaining fair results under different conditions.

The organization of the paper reflects our point of view about which are the main contributions of the paper.
The paper starts with a general theoretical contribution, that is later refined for a specified policy family and then we propose two algorithms for determining the step size when the policy gradient is estimated through samples. Finally, we thought that adding some empirical results, even on a toy example, is actually relevant. Since the algorithms are straightforward derived by the theoretical results, we thought that the current paper organization was quite smooth. Nevertheless, we will consider the reviewer's suggestion for the final version.

Assigned_Reviewer_8

We would like to thank the reviewer for providing several detailed comments to improve our paper.

The numbers in Table 2 are the number of updates of policy parameters (like in Table 1).

We think that the other issues raised by the reviewer have been clarified by the arguments reported at the beginning.


Assigned_Reviewer_9

The contributions of this paper are mainly theoretical, but we think that they can be a strong basis to produce practical algorithms.
Results in Table 2 have the goal to empirically evaluate the bounds derived in Section 5.
As noted by the reviewer, results show that even with a quite large number of trajectories the bounds on the gradient are still not small enough to get really close to the optimal policy while producing a sequence of monotonically improving policies.
Empirical results are meant to evaluate both strengths and weaknesses of the proposed approaches, and the latter ones are helpful to suggest future research directions (in this case identification of tighter bounds will clearly improve the practical implications of the proposed approach).

We think that the other issues raised by the reviewer have been clarified by the arguments reported at the beginning.